# Histone serotonylation in dorsal raphe nucleus contributes to stress- and antidepressant-mediated gene expression and behavior

Mood disorders are an enigmatic class of debilitating illnesses that affect millions of individuals worldwide. While chronic stress clearly increases incidence levels of mood disorders, including major depressive disorder (MDD), stress-mediated disruptions in brain function that precipitate these illnesses remain largely elusive. Serotonin-associated antidepressants (ADs) remain the first line of therapy for many with depressive symptoms, yet low remission rates and delays between treatment and symptomatic alleviation have prompted skepticism regarding direct roles for serotonin in the precipitation and treatment of affective disorders. Our group recently demonstrated that serotonin epigenetically modifies histone proteins (H3K4me3Q5ser) to regulate transcriptional permissiveness in brain. However, this non-canonical phenomenon has not yet been explored following stress and/or AD exposures. Here, we employed a combination of genome-wide and biochemical analyses in dorsal raphe nucleus (DRN) of male and female mice exposed to chronic social defeat stress, as well as in DRN of human MDD patients, to examine the impact of stress exposures/MDD diagnosis on H3K4me3Q5ser dynamics, as well as associations between the mark and depression-related gene expression. We additionally assessed stress-induced/MDD-associated regulation of H3K4me3Q5ser following AD exposures, and employed viral-mediated gene therapy in mice to reduce H3K4me3Q5ser levels in DRN and examine its impact on stress-associated gene expression and behavior. We found that H3K4me3Q5ser plays important roles in stress-mediated transcriptional plasticity. Chronically stressed mice displayed dysregulated H3K4me3Q5ser dynamics in DRN, with both AD- and viral-mediated disruption of these dynamics proving sufficient to attenuate stress-mediated gene expression and behavior. Corresponding patterns of H3K4me3Q5ser regulation were observed in MDD subjects on vs. off ADs at their time of death. These findings thus establish a neurotransmission-independent role for serotonin in stress-/AD-associated transcriptional and behavioral plasticity, observations of which may be of clinical relevance to human MDD and its treatment.

✉e-mail: ian.maze@mssm.edu

Major depressive disorder (MDD), along with related mood disorders, is an enigmatic and highly heterogeneous syndrome that affects approximately 17 million American adults each year[1]. Chronic stress exposures represent a major risk factor for MDD[2], however the molecular mechanisms underlying stress-induced susceptibility to depression remain poorly understood. Despite being serendipitously discovered more than 60 years ago, antidepressant (AD) treatments that target monoaminergic systems (e.g., selective serotonin reuptake inhibitors/SSRIs) remain the first line of therapy for many with MDD. Yet long delays between initiation of treatment and symptomatic alleviation, along with low remission rates[3], have encouraged further investigation to identify more direct therapeutic targets. The monoamine neurotransmitter serotonin, or 5-hydroxytryptamine (5-HT), in particular, is thought to play critical roles in neuronal plasticity associated with affective disorders, as altered serotonergic signaling is implicated in both the etiology and treatment of MDD[4]. However, a recent report revealing a lack of robust evidence linking alterations in serotonin levels to MDD and AD efficacy has prompted renewed interest from the field in defining precise roles for 5-HT in the precipitation and treatment of MDD[5].

In the central nervous system, 5-HT has long been thought to function primarily as a neuromodulator, regulating a wide array of physiological and behavioral functions, including cognitive and emotional processing, autonomic control and sleep-wake cycles[6]. In the brain, 5-HT is synthesized predominantly in monoaminergic, tryptophan hydroxylase 2-expressing neurons located in the dorsal raphe nucleus (DRN). 5-HT is thought to elicit its neuromodulatory effects via a complex and wide-ranging efferent system that projects broadly throughout the brain (including to key regions of the limbic system, such as the prefrontal cortex, nucleus accumbens and amygdala, as well as the hippocampus and cerebellum, among others)[7]. In this well-documented view, 5-HT receptor-mediated mechanisms initiate alterations in cell-cell communication, which in turn can contribute to the plasticity of postsynaptic neurons[8–10]. During early brain development, 5-HT can additionally act as a trophic factor to regulate neuronal growth and differentiation processes, synaptogenesis and dendritic pruning[11–13], suggesting potential roles for this molecule beyond its actions as a neurotransmitter. Along these lines, while SSRIs function pharmacologically to perturb 5-HT signaling in brain via inhibition of the 5-HT transporter (SLC6A4/SERT)—a phenomenon that contributed to the development of the 'monoamine hypothesis of depression'[14]—it remains unclear whether serotonergic dysfunction itself promotes MDD-related pathologies, or how therapeutics might work mechanistically to promote symptomatic alleviation in MDD individuals.

While vesicular packaging of monoamines is essential for neurotransmission, previous data demonstrated the additional presence of extravesicular monoamines in both the soma and nucleus of monoaminergic neurons[15,16]. In addition to its role as a neuromodulator, 5-HT was previously shown to be capable of forming covalent bonds with certain substrate proteins via transamidation by the tissue Transglutaminase 2 enzyme, a process referred to as serotonylation[17]. In more recent studies, our group identified a new class of histone post-translational modification (PTM) termed monoaminylation, whereby monoamine neurotransmitters, such as 5-HT, dopamine and histamine, can be transamidated onto histone glutamine residues[18–23]. We showed that histone H3 glutamine (Q) 5 is a primary site for these PTMs and demonstrated that H3 monoaminylation states play important roles in the regulation of neuronal transcriptional programs, both during early development/cellular differentiation and in adult brain. We demonstrated that combinatorial H3 lysine 4 (K4) tri-methylation (me3) glutamine 5 (Q5) serotonylation (H3K4me3Q5ser), in particular, acts as a permissive epigenetic mark, both by enhancing the binding of the general transcription factor complex TFIID, and attenuating

H3K4me3 demethylation via inhibition of K4me3 demethylases[18,24]. While these PTMs play critical roles in the regulation of normal patterns of transcription in brain, we also found that certain H3 monoaminylations (e.g., H3 dopaminylation) are inappropriately dynamic in response to aberrant environmental stimuli, which contribute to maladaptive neuronal plasticity in disorders associated with altered monoaminergic signaling (e.g., cocaine and opiate use disorders)[19,25,26].

Given the chronic, relapsing nature of MDD, great efforts have been taken over the past two decades to examine the underlying molecular determinants of this brain disorder, the findings of which have uncovered various patterns of transcriptional and epigenetic dysregulation—often brain region and cell-type specific—as potential causative factors in the precipitation and persistence of MDD-related pathophysiology[27,28]. Furthermore, explorations in preclinical rodent models of chronic stress, which can be used to model specific endophenotypes associated with MDD (e.g., anhedonia, social avoidance, behavioral despair, cognitive deficits, etc.), have revealed strong correlations between epigenetic dysfunction, gene expression abnormalities and behavioral stress susceptibility[29–34]. However, our understanding of how these mechanisms mediate life-long susceptibility to stress-induced syndromes like MDD remains limited. Additionally, while much evidence exists implicating molecular alterations in cortical and limbic brain structures (all of which receive dense serotonergic projections) as precipitating factors in the regulation of stress susceptibility vs. resilience[35–40], fewer studies have explored chromatin-related phenomena in DRN, which may also contribute significantly to behavioral dysregulation in affect-related disorders.

Here, we demonstrate that DRN displays significant alterations in mood disorder-associated gene expression programs following chronic social stress in both male and female mice, with behaviorally resilient vs. susceptible animals displaying blunted transcriptional abnormalities. We further show that histone H3 serotonylation patterns are reorganized in response to chronic stress in both sexes, a phenomenon that is rescued in both behaviorally resilient animals and mice chronically treated with the SSRI fluoxetine. Finally, we demonstrate that directly reducing levels of H3 serotonylation in DRN using a dominant negative viral vector approach is sufficient to reverse chronic stress-induced gene expression programs and promote behavioral resilience to stressful stimuli. In sum, these findings establish a non-canonical, neurotransmission-independent role for 5-HT in stress-mediated transcriptional and behavioral plasticity in DRN, and indicate that certain ADs may function, at least in part, to reverse altered patterns of H3 serotonylation in brain.

## Results

### Gene expression programs in DRN are responsive to chronic social stress

To begin investigating the impact of chronic stress exposures on gene expression programs in DRN, we performed chronic social defeat stress (CSDS) in adult male mice, a well-characterized and etiologically relevant rodent model for the study of human depression, which recapitulates numerous pathophysiological features of MDD (e.g., social avoidance, anhedonia, stress-related metabolic syndromes, etc.) and displays symptomatic reversal in response to chronic, but not acute, AD treatments[36,41,42]. CSDS in male mice produced two distinct groups of stress-susceptible vs. stress-resilient animals, with stress-susceptible mice displaying heightened levels of social avoidance in comparison to resilient and control (i.e., non-stressed, handled) groups (Fig. 1A). We then performed bulk RNA-seq on DRN tissues from control vs. susceptible vs. resilient mice, followed by differential expression analysis to compare the three groups. We found that stress-susceptible male mice exhibited significant alterations in the expression of 2266 protein-coding genes (PCGs; FDR < 0.1) vs. respective handled controls. Subsequent unsupervised clustering of all three

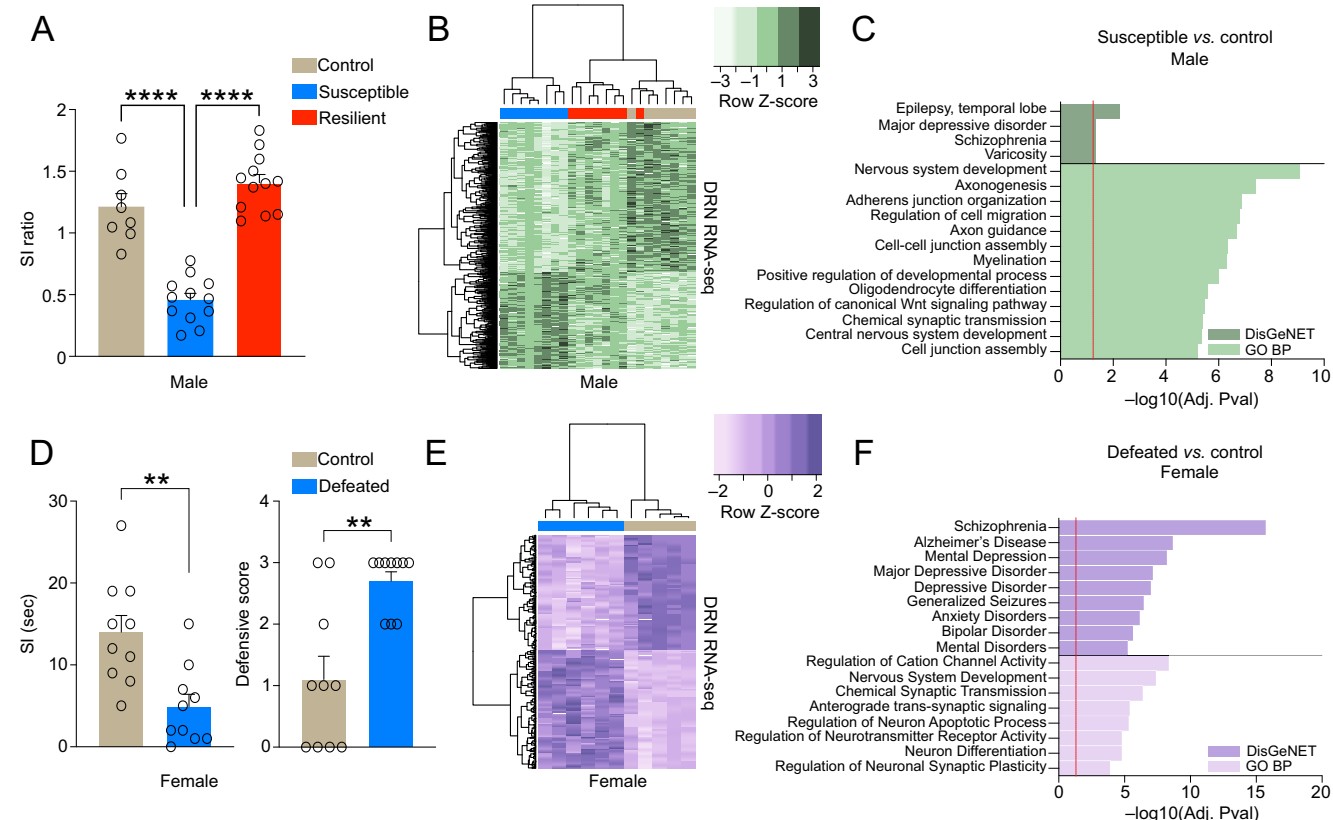

**Fig. 1 | Chronic social stress in both male and female mice results in altered gene expression in DRN. A** SI ratio of control vs. stress-susceptible vs. stress-resilient male mice ($n = 8$ for controls, 12 for susceptible and 12 for resilient groups). One-way ANOVA significant main effects observed ($p < 0.0001$, $F_{2,29} = 53.44$). Tukey's MC test: control vs. susceptible mice ($p < 0.0001$) and susceptible vs. resilient mice ($p < 0.0001$). **B** Clustering of control, susceptible and resilient groups for 1502 differentially expressed (DE) genes (susceptible vs. control; $n = 7-8$/group, FDR < 0.05). **C** Pathway enrichment (FDR < 0.05; Benjamini–Hochberg) for the PCGs differentially expressed (FDR < 0.1) in susceptible vs. control males. Dashed line indicates significance via adjusted p value. **D**$_{Left}$ SI time of control vs. socially

defeated female mice ($n = 10$/group). Student's two-tailed $t$ tests: defeated vs. control mice ($p = 0.0021$, $t_{18} = 3.582$). **D**$_{Right}$ Defensive scores for control vs. socially defeated female mice. Mann-Whitney test (unpaired): defeated vs. control mice ($p = 0.0034$, $U = 14.50$). **E** Clustering of defeated and control groups for 234 DE genes (defeat vs. control; $n = 5-6$/group, FDR < 0.05). **F** Pathway enrichment (FDR < 0.05; Benjamini–Hochberg) for PCGs differentially expressed in defeat vs. control females (at FDR < 0.1). Dashed line indicates significance via adjusted p value. For bar graphs, data presented as mean ± SEM. Source data are provided as a Source Data file.

groups at these differentially regulated transcripts revealed a clear pattern of separation between stress-susceptible vs. control animals, with resilient mice displaying a pattern more similar to that of controls (only 56 PCGs were found to be differentially regulated comparing resilient vs. control mice; FDR < 0.1) (Fig. 1B, Supplementary Data 1, 8, 9). Subsequent gene set enrichment analyses on dysregulated loci observed in susceptible vs. control mice [GO Biological Process and DisGeNET, the latter of which curates a collection of genes and variants associated with human diseases, integrating data from publicly available genomics repositories, GWAS catalog, animals models (focused on genotype x phenotype relationships) and current scientific literature] identified significant (FDR < 0.05) overlaps with pathways/processes involved in neuronal development (e.g., axon guidance, axonogenesis, regulation of cell migration, etc.) and synaptic transmission (e.g., chemical synaptic transmission), as well as enrichment in disease associated pathways related to psychiatric and affect-related disorders (e.g., MDD and schizophrenia) (Fig. 1C, Supplementary Data 2, 3). Interestingly, while both up- (988) and downregulated (1278) gene expression was observed in stress-susceptible male DRN vs. controls, downregulated genes appear to have contributed more significantly to gene ontology and disease pathway enrichment observed in Fig. 1C (FDR < 0.05; Supplementary Fig. 1A, B, Supplementary Data 4–7). These findings suggest that stress-susceptible gene expression programs in male DRN, particularly those genes that are

acutely repressed in response to chronic stress, may be relevant to aberrant patterns of neuronal and/or behavioral plasticity observed in response to chronic stress exposures.

Given a vast literature indicating prominent sex differences with respect to disparities of onset, lifetime prevalence and symptomatic presentation of MDD in humans[43–45], as well as stress vulnerability phenotypes in preclinical animal models[46], we next sought to examine gene expression programs in DRN of chronically stressed (i.e., defeated) female mice in order to compare to those transcriptional patterns observed in males. To do so, we performed a recently established CSDS paradigm in female mice that similarly recapitulates numerous features of MDD, as well as behaviors observed in male rodents following CSDS[47,48], including increased social avoidance and heightened levels of defensive behaviors (Fig. 1D). In our paradigm, defeated females were found to be entirely susceptible to CSDS, so we therefore compared defeated females to susceptible males in subsequent analyses. Following CSDS, we again performed bulk RNA-seq on DRN tissues from control vs. defeated female mice, followed by differential expression analysis to compare the two groups. We found that defeated females exhibited significant alterations in the expression of 339 PCGs (FDR < 0.1) vs. respective controls—far fewer than that observed in males following CSDS—with unsupervised clustering revealing a clear pattern of separation between subjects by 'treatment' type (Fig. 1E, Supplementary Data 10). While a more modest number of

loci were found to be dysregulated in female vs. male DRN, gene set enrichment analyses (GO Biological Process and DisGeNET) again identified significant (FDR < 0.05) overlaps with shared pathways/processes involved in nervous system development, chemical synaptic transmission and psychiatric/mood-related disorders (e.g., MDD depressive disorder, anxiety disorders, bipolar disorder, etc.) (Fig. 1F, Supplementary Data 11, 12). Again, downregulated genes in females were found to contribute most significantly to gene ontologies observed when assessing the dataset irrespective of directionality (FDR < 0.05; Supplementary Fig. 1C, D, Supplementary Data 13–16). In addition, while male susceptible mice clearly displayed more robust alterations in gene expression vs. defeated females, a subset of these dysregulated loci were found to significantly overlap between the sexes (odds ratio = 3.1; p = 1.3e-16) (Supplementary Fig. 1E), with these shared genes also displaying significant enrichment for pathways/processes involved in brain development, synaptic transmission and mood-related disorders (FDR < 0.05; Supplementary Fig. 1F, Supplementary Data 17, 18). These data indicate that, as in stress-susceptible males, gene expression programs in defeated female DRN also appear relevant to abnormal neuronal and behavioral plasticity associated with affective disturbances.

### H3 serotonylation is altered genome-wide in DRN of male and female mice acutely following chronic social stress

Given the transcriptional responsiveness of DRN to chronic stress exposures in both male and female mice, we next sought to interrogate potential chromatin-related mechanisms that may contribute to these observed dynamics. Since DRN is enriched for 5-HT-producing neurons, a monoaminergic cell population that also displays robust enrichment for histone H3 serotonylation[18], we further explored potential regulation of H3K4me3Q5ser dynamics in DRN of mice 24 hr following social interaction (SI) testing. Using western blotting to first assess global levels of the combinatorial PTM across all three groups of controls and stress-susceptible vs. stress-resilient mice (one-way ANOVA), we observed robust differences between stress-susceptible vs. stress-resilient animals, with stress-susceptible mice displaying significant deficits in the mark (Fig. 2A). While not significant (p = 0.1204), stress-susceptible mice also displayed nominal reductions in the mark compared to non-stressed controls. Similarly, when comparing female defeated vs. control animals 24 hr after CSDS, we found that the serotonylation mark was also downregulated (Fig. 2B). To assess whether these changes may be clinically relevant, we next measured H3K4me3Q5ser levels in postmortem DRN of humans with MDD vs. demographically matched controls, where we found that the serotonylation mark was also downregulated in MDD individuals without ADs present at their time of death (Fig. 2C, left; p < 0.05, unpaired t test). Interestingly, however, when comparing levels of the mark in DRN from humans diagnosed with MDD with ADs present at their time of death, we observed that H3K4me3Q5ser levels were similar to those of their respectively matched controls (Fig. 2C, right; p > 0.05, unpaired t test), suggesting a potential interaction between the mark's expression, MDD diagnosis and AD exposures. A potential caveat to the aforementioned human tissue analysis is the limited cohort size under investigation, which has resulted from difficulties in obtaining larger numbers of human postmortem DRN tissues from individuals diagnosed with MDD. In future efforts, it will therefore be important to confirm these western blotting results with larger numbers of control vs. MDD DRN samples.

Next, to assess whether alterations in global levels of H3K4me3Q5ser correspond with meaningful patterns of genomic regulation following chronic stress exposures, we performed ChIP-seq for the mark in DRN of both males (control vs. susceptible vs. resilient) and females (control vs. defeated) 24 hr after SI testing. Following peak calling (FDR < 0.05, >5 fold-enrichment over input, Supplementary Data 19–23), we first assessed the degree of overlap between PCGs

enriched for H3K4me3Q5ser in control males vs. females, where we observed a significant degree of overlap between the sexes (odds ratio = 79.4; p = 0e + 00); Fig. 2D), with non-overlapping peaks largely representing sex-specific loci (e.g., *Xist* in females and *Kdm5d* in males; Fig. 2E). Note that the majority of peaks identified in both male and female DRN were found to be located throughout genic loci, particularly within gene promoters and often enriched proximally to transcriptional start sites (Supplementary Fig. 2A, B). We then performed differential enrichment analysis for the mark (FDR < 0.05, Log2FC ≥ 1.5 or ≤ −1.5, Supplementary Data 24, 27) at peak-enriched PCGs (the most common sites of differential enrichment in both males and females; Supplementary Fig. 2C, D) comparing the degree of overlap between differentially enriched loci between stress-susceptible males vs. defeated females (vs. their respective controls), where we observed that ~42.1% and ~48.6% of differentially enriched PCGs overlapped (odds ratio = 8.2; p = 0e + 00) between males and females, respectively, following chronic social stress (Fig. 2F). Similar to the gene expression results presented in Fig. 1, these overlapping gene sets between male and female stress-exposed mice were demonstrated to enrich (FDR < 0.05) for pathways/biological processes (GO Biological Process) associated with neuronal development and synaptic regulation, as well as disease-enriched (GWAS catalog, DisGeNET) loci related to psychiatric, neurodevelopmental and affect-related disorders (e.g., MDD, Irritable Mood, Feeling Worry, Bipolar Depression, Unipolar Depression, etc.) (Fig. 2G, Supplementary Data 28–30). Importantly, a subset of these differentially enriched loci were observed to significantly overlap with genes demonstrated to be differentially expressed in response to stress in both males (odds ratio = 2.0; p = 1.1e-27) and females (odds ratio = 1.8; p = 4.5e-07) (Supplementary Fig. 2E, F), with these overlapping PCGs also displaying significant enrichment (FDR < 0.05) for disease pathways (DisGeNET) associated with MDD, mood disorders, bipolar disorder, unipolar depression, etc. in both sexes (Supplementary Fig. 2G, Supplementary Data 34, 35). Finally, given our earlier western blotting results in males showing that decreased H3K4me3Q5ser levels in susceptible animals were not observed in resilient mice, we next examined the degree of overlap (odds ratio = 10.1; p = 0e + 00) between differentially enriched PCGs in stress-susceptible vs. stress-resilient mice, finding that ~90.6% of PCGs exhibiting dynamics in susceptible mice displayed reversal in these enrichment patterns in stress-resilient animals (Fig. 2H, Supplementary Data 25, 26). Subsequent gene set enrichment analyses (FDR < 0.5) again revealed strong associations between those loci displaying reversals in enrichment between susceptible vs. resilient mice and pathways/processes (GO Biological process) related to neurodevelopmental processes and synaptic organization/function, along with significant enrichment in disease associated pathways (GWAS catalog, DisGeNET) related to affective disorders (e.g., Depression, Feeling Worry, MDD, Bipolar Depression, etc.) and other psychiatric syndromes (Fig. 2I, Supplementary Data 31–33). In sum, our genomics data acutely following CSDS indicated that alterations in H3K4me3Q5ser enrichment patterns in DRN in response to chronic stress significantly correlate with abnormal transcriptional programs associated with MDD and other affective disorders.

### Chronic AD treatments reverse stress susceptibility and rescue stress-induced H3 serotonylation dynamics in DRN

Considering that our western blotting data in human DRN revealed that global levels of H3K4me3Q5ser were altered in individuals with MDD without ADs onboard at their time of death, an effect that was not observed in patients with ADs onboard at their time of death vs. respectively matched controls, we next sought to explore whether the mark may similarly be responsive to chronic AD treatments following CSDS in mice. To examine this, male mice were subjected to 10 days of CSDS, assessed for SI and separated into control vs. susceptible vs. resilient populations (Fig. 3A–C; Pre-treatment) before being treated

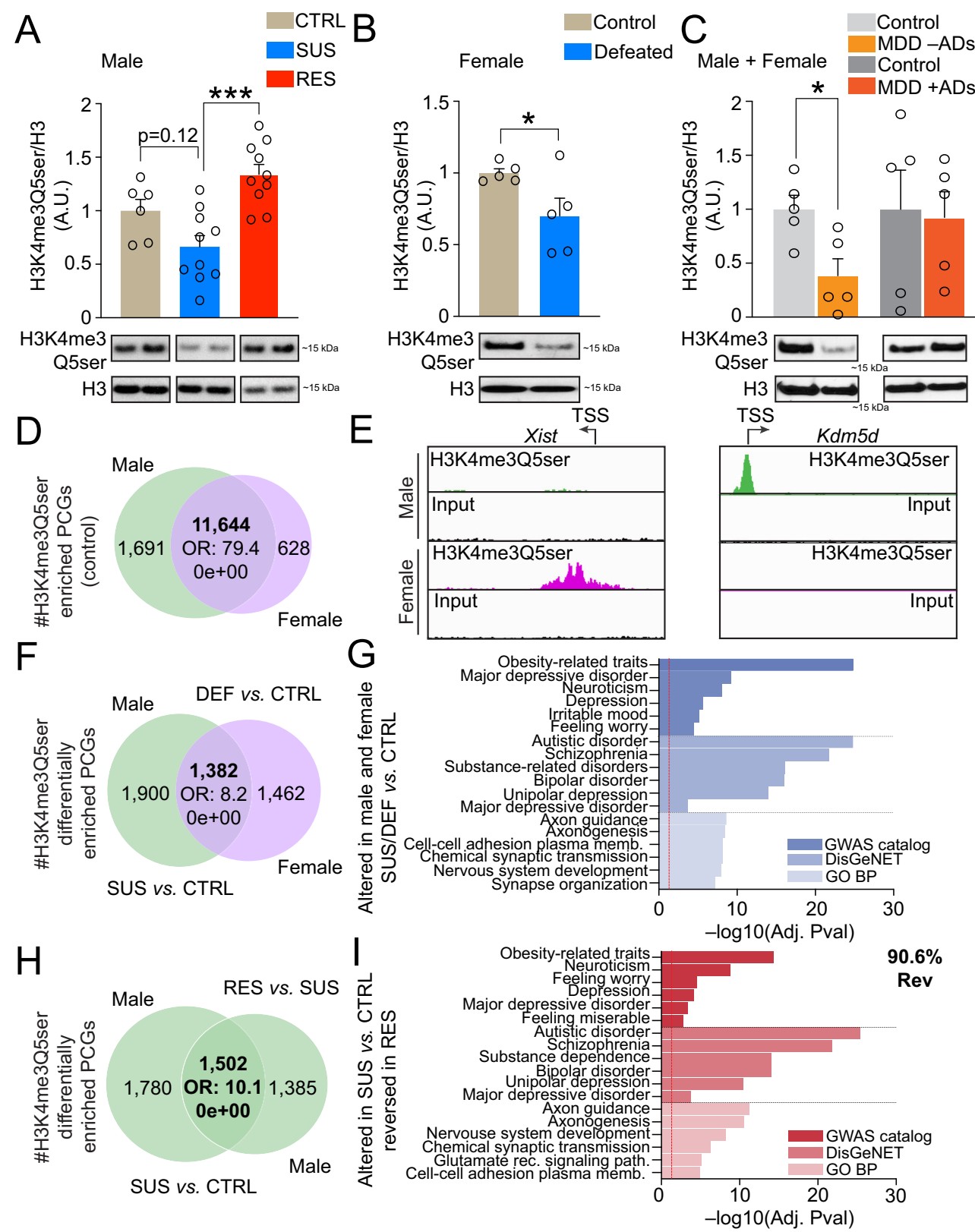

for 30 days with the SSRI AD fluoxetine vs. water as a vehicle control[49] (Fig. 3A–C; Post-treatment). Following another round of SI testing to examine behavioral reversal of the susceptibility phenotype in previously susceptible mice, DRN tissues were collected for western blotting analysis of H3K4me3Q5ser. As expected, susceptible mice remained susceptible, as measured via SI, following chronic treatments with water (Fig. 3B). However, susceptible animals treated with

chronic fluoxetine displayed significant reversal of previously observed SI deficits (Fig. 3C). Using this protracted timeline, which may better reflect the persistence of stress-vulnerable states vs. examinations 24 hr post-CSDS (as in Fig. 2), we no longer observed a trend toward a global downregulation of H3K4me3Q5ser—a phenomenon that was seen one-day following chronic stress in susceptible animals—but rather found that the mark significantly accumulates in

**Fig. 2 | Chronic social stress promotes altered H3 serotonylation dynamics in DRN. A** H3K4me3Q5ser in DRN of control ($n = 6$) vs. stress-susceptible ($n = 10$) vs. stress-resilient male mice ($n = 10$). One-way ANOVA significant main effects observed ($p = 0.0002$, $F_{2,23} = 12.43$). Tukey's MC test: susceptible vs. resilient mice ($p = 0.0001$). **B** H3K4me3Q5ser in DRN of control vs. defeated female mice ($n = 5$/group). Student's two-tailed $t$ test: defeated vs. control mice ($p = 0.0473$, $t_8 = 2.341$). **C** H3K4me3Q5ser in DRN from human postmortem brain of MDD individuals ± antidepressants onboard at time of death vs. respective controls ($n = 5$/group). Student's two-tailed $t$ tests (individual MDD groups vs. matched controls): MDD −AD's vs. controls ($p = 0.0166$, $t_8 = 3.020$). For western blotting graphs, *$p < 0.05$, ***$p < 0.001$. A.U., arbitrary units, normalized to controls; total histone H3 levels were used as loading controls. **D** Overlap between H3K4me3Q5ser enriched PCGs (FDR < 0.05; Fisher's exact test) in control male vs. female DRN ($n = 3$/group, 3–4 samples pooled per $n$). Odds ratio (OR) and respective p value of overlap are provided. **E** IGV tracks for two sex-specific loci displaying sex-specific enrichment

of permissive H3K4me3Q5ser vs. respective inputs. **F** Overlap between male vs. female PCGs displaying differential enrichment for H3K4me3Q5ser in DRN as a consequence of CSDS [male susceptible vs. control, and female defeated vs. control; $n = 3$/group, 3–4 samples pooled per $n$, FDR < 0.05 (Fisher's exact test)]. OR and respective p value of overlap is provided. **G** Pathway enrichment for PCGs displaying overlapping (male vs. female; 1382 PCGs) differential enrichment for H3K4me3Q5ser as a consequence of CSDS (FDR < 0.05; Benjamini−Hochberg). **H** Overlap between male susceptible vs. control and male resilient vs. susceptible PCGs displaying altered H3K4me3Q5ser enrichment in DRN [$n = 3$/group, 3–4 samples pooled per $n$, FDR < 0.05 (Fisher's exact test)]. OR and respective p value of overlap are provided. **I** Pathway enrichment for PCGs displaying overlapping and reversed differential enrichment for H3K4me3Q5ser in male susceptible vs. control and male resilient vs. susceptible comparisons (FDR < 0.05; Benjamini−Hochberg). See Supplementary Fig. 7A−C for uncropped blots. Data presented as mean ± SEM. Source data are provided as a Source Data file.

DRN of stress-susceptible mice treated with water vs. vehicle treated controls and stress-resilient animals (Fig. 3D). This accumulation, however, was found to be significantly attenuated by chronic fluoxetine treatments in stress-susceptible mice, with levels of the mark normalizing to those of both control and resilient animals; fluoxetine administration did not impact levels of the mark in control or resilient mice, animals that remained behaviorally unaffected in response to chronic AD treatments. These data demonstrated that behavioral responsiveness to ADs following chronic stress in susceptible mice (but not in the absence of stress or in resilient animals) corresponds with reductions in H3K4me3Q5ser levels in DRN, perhaps suggesting a role for AD-mediated H3K4me3Q5ser downregulation in the alleviation of stress-induced behavioral deficits.

Given these western blotting results, we next aimed to explore H3K4me3Q5ser dynamics genome-wide at this protracted timepoint following CSDS −/+ chronic fluoxetine treatments. Following ChIP-seq for the mark in DRN tissues from male control vs. stress-susceptible mice−vehicle ($H_2O$) vs. fluoxetine−we first assessed the degree of overlap between differentially enriched PCGs regulated by chronic stress at acute (stress-susceptible vs. control, 24 hr post-CSDS) vs. protracted (stress-susceptible $H_2O$ vs. control $H_2O$, 30 d post-CSDS) periods following CSDS. While we found that a greater amount of genes displayed stress-induced H3K4me3Q5ser dynamics (FDR < 0.05, Log2FC ≥ 1.0 or ≤ −1.0; note that a slightly lower Log2FC cutoff was used in these comparisons vs. those in Fig. 2 to account for batch variability between experiments) at acute (3879) vs. protracted (718) periods following CSDS in the absence of fluoxetine (the majority of which for both comparisons displayed increased enrichment of the mark), 223 of these PCGs were found to significantly overlap between the two timepoints (odds ratio = 2.3; $p = 1.4e-21$); Supplementary Fig. 3A, Supplementary Data 36−39), with many of these overlapping genes (75%) displaying consistent patterns of regulation by CSDS (127 up/up, 41 down/down). Furthermore, gene set enrichment analyses (FDR < 0.5) of these 223 overlapping PCGs revealed strong associations with pathways/processes (GO Biological process) related to synaptic organization/function, along with significant enrichment in disease associated pathways (GWAS catalog) related to affective disorders (e.g., MDD, depressive symptoms) and other psychiatric syndromes (Supplementary Fig. 3B, Supplementary Data 40, 41). We next aimed to assess the impact of chronic fluoxetine exposures, which effectively reversed stress-susceptibility and stress-induced gene expression (see Fig. 3B, C, Supplementary Fig. 3C, Supplementary Data 42, 43), on H3K4me3Q5ser dynamics genome-wide in mouse DRN. In doing so, we found that 81% of overlapping PCGs (comparing susceptible $H_2O$ vs. control $H_2O$ and susceptible fluoxetine vs. susceptible $H_2O$ gene lists) displaying differential enrichment of the mark at protracted periods following CSDS exhibited restoration of these dynamics in response to

chronic fluoxetine treatments (odds ratio = 8.6; $p = 1.4e-162$), with fluoxetine exposures additionally resulting in a robust loss of H3K4me3Q5ser enrichment at a large number of genes (3495) that were not observed to be regulated in their serotonylation state by chronic stress alone (i.e., an apparent interaction between stress x fluoxetine was clearly observed; Fig. 3E−G, Supplementary Data 44−46); note that many fewer PCGs were found to display significant H3K4me3Q5ser dynamics in DRN as a consequence of chronic fluoxetine exposures in control mice, which is consistent with a lack of behavioral responsiveness to ADs in these animals (Supplementary Data 47). Again, gene set enrichment analyses (FDR < 0.05) of PCGs displaying altered enrichment in stress-susceptible fluoxetine vs. stress-susceptible $H_2O$ mice revealed strong associations with pathways/processes (GO Biological process) related to synaptic organization/function, along with significant enrichment in disease associated pathways (GWAS catalog) related to mood disorders (e.g., MDD, bipolar disorder, mood instability) and other psychiatric syndromes (Fig. 3H, Supplementary Data 48, 49). These data suggested that while fluoxetine indeed functions, at least in part, to reverse stress-induced H3K4me3Q5ser dynamics in DRN, it also serves to promote more global alterations (predominantly reduced enrichment) of the mark, which may additionally contribute to reversals in stress-susceptibility. Next, given that >90% of overlapping stress-regulated genes between stress-resilient vs. stress-susceptible mice 24 hr post-CSDS were found to display opposing patterns of H3K4me3Q5ser regulation (Fig. 2H), we next sought to assess whether PCGs displaying altered H3K4me3Q5ser enrichment in stress-susceptible mice + chronic fluoxetine may overlap with PCGs displaying reversals in the mark's enrichment in stress-resilient vs. stress-susceptible comparisons (FDR < 0.05, Log2FC ≥ 1.0 or ≤ −1.0, Supplementary Data 50). Indeed, we identified significant overlaps between these two comparisons (Supplementary Fig. 3D, odds ratio = 2.5; $p = 4.8e-90$), with the overlapping genes significantly enriching (FDR < 0.05) for pathways/processes (GO Biological process) related to synaptic organization/function, along with significant enrichment in disease associated pathways (GWAS catalog) related to mood disorders (e.g., MDD, bipolar disorder, positive affect) and other psychiatric illnesses (Supplementary Fig. 3E, Supplementary Data 51, 52). Finally, to examine whether fluoxetine-induced changes in H3K4me3Q5ser enrichment that were observed in stress-susceptible mice correlate with genes displaying similar patterns of regulation in human brain, we next performed ChIP-seq for the mark in human postmortem DRN tissues from individuals diagnosed with MDD −/+ ADs onboard at their time of death (FDR < 0.05, Log2FC ≥ 1.0 or ≤ −1.0, Supplementary Data 53−58). While only a handful of differentially enriched PCGs were observed when comparing MDD −ADs vs. matched controls−likely owing to the heterogenous nature of MDD and our limited sample size−such

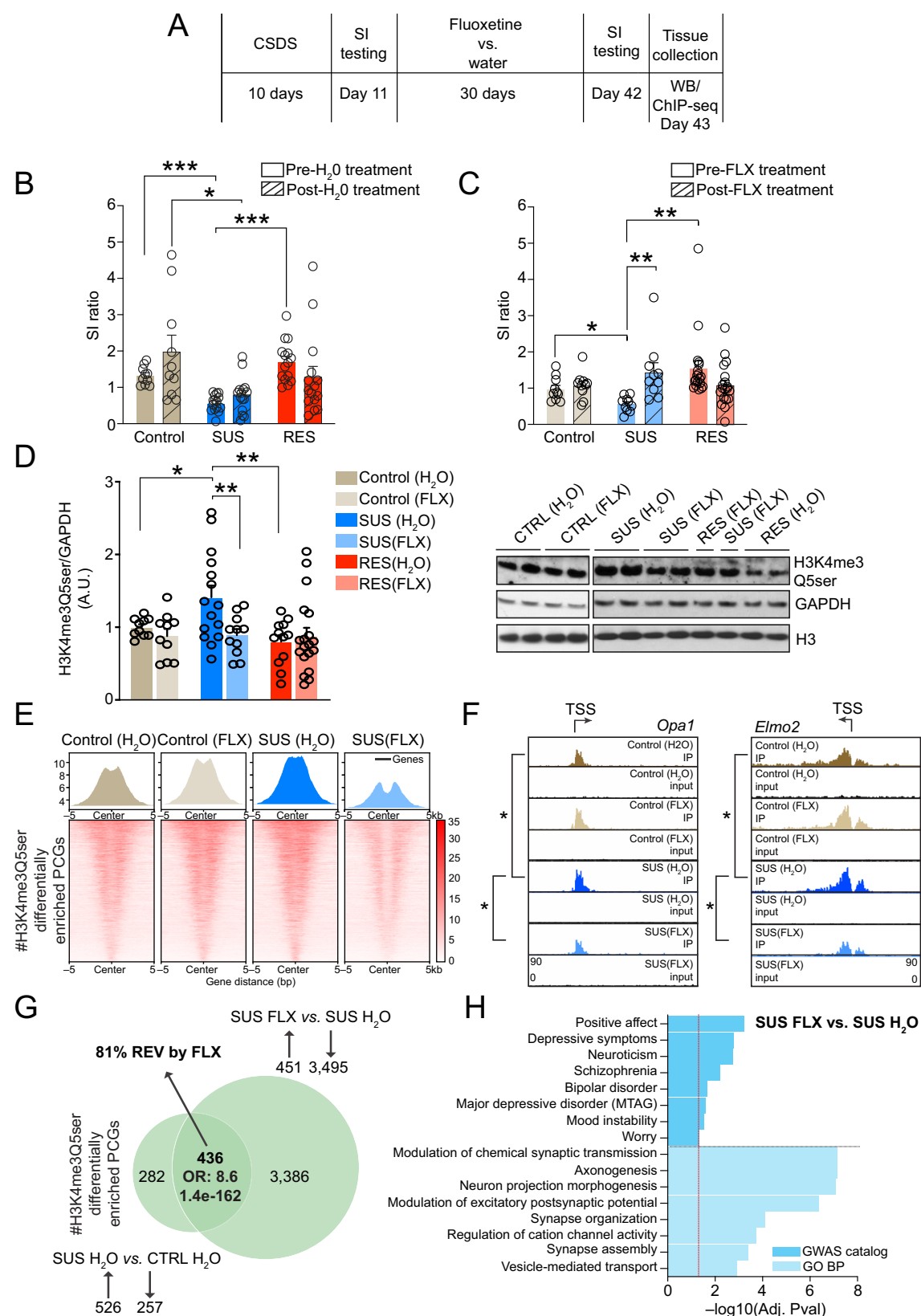

assessments did identify significant and ontologically relevant (FDR < 0.05, Supplementary Data 59–60) overlaps between PCGs displaying altered dynamics of the mark in susceptible fluoxetine vs. susceptible $H_2O$ mice and in human subjects with MDD + vs. −ADs (Supplementary Fig. 3F–G, odds ratio = 2.2; p = 4.4e-92); note that a greater number of PCGs in total displayed loss (4146) vs. gain (3034) of the mark in MDD

patients + vs. −ADs, data which are consistent with our fluoxetine findings in mice. These data suggest that alterations in H3K4me3Q5ser enrichment observed in behaviorally responsive, fluoxetine treated CSDS mice may be clinically relevant and may reflect functionally important chromatin adaptations that occur in human MDD subjects undergoing AD treatments.

**Fig. 3 | Chronic fluoxetine treatments rescue behavioral deficits and stress-induced H3K4me3Q5ser dynamics in DRN in stress-susceptible male mice.**
**A** Timeline: fluoxetine vs. water experiments. **B** SI ratio: control ($n = 10$), stress-susceptible ($n = 15$) and stress-resilient ($n = 15$), pre- vs. post-30 days water. Two-way RM ANOVA: stress ($p = 0.0005$, $F_{2,37} = 9.298$) and stress x time ($p = 0.0234$, $F_{2,37} = 4.162$). Posthoc $t$ tests with Bonferroni correction: control vs. susceptible, pre-treatment ($p = 0.0003$); susceptible vs. resilient, pre-treatment ($p = 0.0003$); and control vs. susceptible, post-treatment ($p = 0.0201$). **C** SI ratio: control ($n = 10$), susceptible ($n = 10$) and resilient ($n = 19$), pre- vs. post-30 days fluoxetine. Two-way RM ANOVA: stress x treatment ($p = 0.0018$, $F_{2,36} = 7.548$). Bonferroni's MC tests: susceptible mice, pre- vs. post-30 days fluoxetine ($p = 0.0098$). Posthoc $t$ tests with Bonferroni correction: control vs. susceptible, pre-treatment ($p = 0.0111$), and susceptible vs. resilient, pre-treatment ($p = 0.0066$). **D** H3K4me3Q5ser in DRN: control ($n = 10$ for water and FLX), susceptible ($n = 15$ water; $n = 11$ FLX) and resilient ($n = 12$ water; $n = 19$ FLX) following 30 days of fluoxetine vs. water. Two-way ANOVA: stress ($p = 0.0289$, $F_{2,71} = 3.725$) and stress x fluoxetine ($p = 0.0420$, $F_{2,71} = 3.316$). Sidak's

MC tests: susceptible post-30 days fluoxetine vs. susceptible post-30 days water ($p = 0.0094$); Tukey's MC: susceptible vs. control, post-30 days water ($p = 0.0554$), and susceptible vs. resilient, post-30 days water ($p = 0.0013$). GAPDH and H3 levels were used as loading controls. **E** H3K4me3Q5ser enrichment at PCGs displaying differential enrichment (FDR < 0.05) between SUS FLX vs. SUS $H_2O$ for each group. **F** IGV tracks for two genes displaying significantly (*diffReps) increased enrichment for H3K4me3Q5ser in SUS ($H_2O$) vs. control ($H_2O$), and rescue in SUS (FLX) vs. SUS ($H_2O$). **G** Overlap between PCGs displaying protracted differential enrichment of H3K4me3Q5ser by CSDS vs. PCGs displaying regulation of the mark by fluoxetine in susceptible mice [$n = 3$/group, 3–4 samples pooled per $n$, FDR < 0.05 (Fisher's exact test)]. OR and respective p values of overlap are provided. **H** Pathway enrichment for PCGs displaying differential enrichment for H3K4me3Q5ser in susceptible FLX vs. susceptible $H_2O$ (FDR < 0.05; Benjamini–Hochberg). *$p < 0.05$, **$p < 0.01$, ***$p < 0.001$. Data presented as mean ± SEM. A.U. arbitrary units; normalized to controls. Supplementary Fig. 7D: uncropped blots. Source data are provided as a Source Data file.

## Directly reducing H3 serotonylation in DRN promotes stress resilience and reversal of stress-mediated gene expression programs

Since we observed that H3K4me3Q5ser levels were elevated in DRN of susceptible vs. resilient mice over protracted periods following chronic stress exposures, a phenomenon that was largely reversed by ADs, we next aimed to explore whether prophylactically reducing H3 serotonylation in DRN may prevent the precipitation of stress-mediated gene expression programs and/or behavioral susceptibility. To examine this, male mice were injected intra-DRN with one of three lentiviral vectors—which transduce both neurons and glia, as all cell-types in DRN have previously been shown to express the serotonylation mark[18]; thus, we aimed to express these constructs in a non-cell-type restrictive manner—expressing either GFP (aka empty) or H3.3 WT controls vs. H3.3Q5A, the latter of which functions as a dominant negative by incorporating into neuronal chromatin without being able to be monoaminylated, thereby reducing levels of H3 serotonylation at affected loci (as demonstrated in primary cultured neurons using a ChIP/re-ChIP-based approach; Supplementary Fig. 4A–D)[18]. A separate cohort of mice were surgerized to validate the efficiency of H3.3 incorporation into neural chromatin in DRN via immunohistochemistry/immunofluorescence, which provided additional validation that expression of H3.3Q5A (vs. H3.3 WT) is sufficient to significantly reduce H3K4me3Q5ser levels by ~48% in transduced cells (Fig. 4A, B). Following viral transduction and recovery, mice underwent CSDS and then were assessed via SI testing to examine avoidance behavior, after which time, virally transduced DRN tissues were collected for RNA-seq analysis (Fig. 4C). Following CSDS in virally transduced animals, we observed significant deficits in social interaction in both viral control groups (empty and H3.3 WT, neither of which impact H3 serotonylation—see Supplementary Fig. 4; note that due to the experimental design of this experiment, susceptible and resilient behavioral readouts occurred post-viral manipulations). However, we found that reducing H3 serotonylation in DRN using the dominant negative H3.3Q5A virus attenuated CSDS-induced social avoidance behavior (in effect increasing the proportion of resilient animals observed post-CSDS in the H3.3Q5A viral group), indicating that viral-mediated downregulation of H3K4me3Q5ser in chronically stressed animals is sufficient to promote behavioral resilience (Fig. 4D). And while our AD data presented in Fig. 3 could not definitively link observed fluoxetine-induced reductions in H3K4me3Q5ser to the reversals of stress susceptibility observed post-AD treatment, those findings were indeed consistent with our viral manipulation experiments, which causally linked inhibition of the mark during stress exposures to the promotion of stress-resilience. Expression of H3.3Q5A in DRN did not affect SI behavior in control (i.e., non-CSDS) mice; however, attenuation of H3 serotonylation in a separate cohort

of non-stressed mice was found to decrease behavioral despair in the forced swim test (FST) (Supplementary Fig. 5A), with no impact of viral manipulations observed in anxiety-related tasks, such as the elevated plus maze (EPM; Supplementary Fig. 5B) or open field test (OFT; Supplementary Fig. 5C).

Next, to examine whether behavioral resilience in H3.3Q5A-expressing mice may correspond to a restoration of gene expression abnormalities elicited by chronic stress exposures, we performed bulk RNA-seq on microdissected, virally transduced DRN tissues from control vs. CSDS mice. Rank-rank hypergeometric overlap (RRHO) analysis revealed, in comparison to gene expression programs potentiated by chronic stress in both control groups (empty—left, H3.3 WT—right), that transduction by H3.3Q5A significantly reversed stress-induced gene expression profiles (Fig. 4E, Supplementary Data 61–64). Importantly, gene expression programs found to be induced by CSDS in virally transduced animals (e.g., empty vector) significantly correlated with differential gene expression patterns observed in susceptible vs. control comparisons (24 hr post-SI testing) using tissues from non-virally transduced mice (from Fig. 2), with H3.3Q5A manipulations similarly reversing the expression of these stress impacted genes (Supplementary Fig. 6A, B). These data demonstrate that H3 serotonylation is important for potentiating stress-associated patterns of transcriptional dysregulation in DRN, abnormalities that may contribute importantly to the behavioral deficits observed. Finally, to elucidate the specific gene sets and biological pathways that may be affected by H3K4me3Q5ser downregulation in stress-susceptible animals, we performed differential expression analysis comparing H3.3Q5A vs. empty-expressing mice −/+ CSDS, and then used the list of significantly rescued genes following H3.3Q5A manipulations (Supplementary Fig. 6C, D) to perform gene ontology analyses (Fig. 4F, Supplementary Data 65–67). These genes were subjected to gene set enrichment analysis (GWAS catalog, DisGeNET and GO Biological process), which significantly implicated phenotypic and disease associations with altered neuronal developmental processes, abnormal emotional/affective behavior, mood disorders and MDD, among others, as being rescued by H3.3Q5A manipulations. Finally, given that both fluoxetine and H3.3Q5A mediated reductions of H3K4me3Q5ser in DRN were sufficient to reverse stress-induced dynamics of the mark and rescue stress-induced gene expression/behavior, we next sought to explore whether these two manipulations might induce alterations at the same genes, thereby linking fluoxetine's genomic and behavioral rescue effects to the impact of directly manipulating the mark in the context of chronic stress. In doing we, we identified significant overlaps (odds ratio = 2.6; $p = 3.6e-146$) between PCGs displaying altered serotonylation dynamics in response to fluoxetine exposures in stress-susceptible mice and genes that were found to be differentially expressed following H3.3Q5A manipulations in the context of CSDS

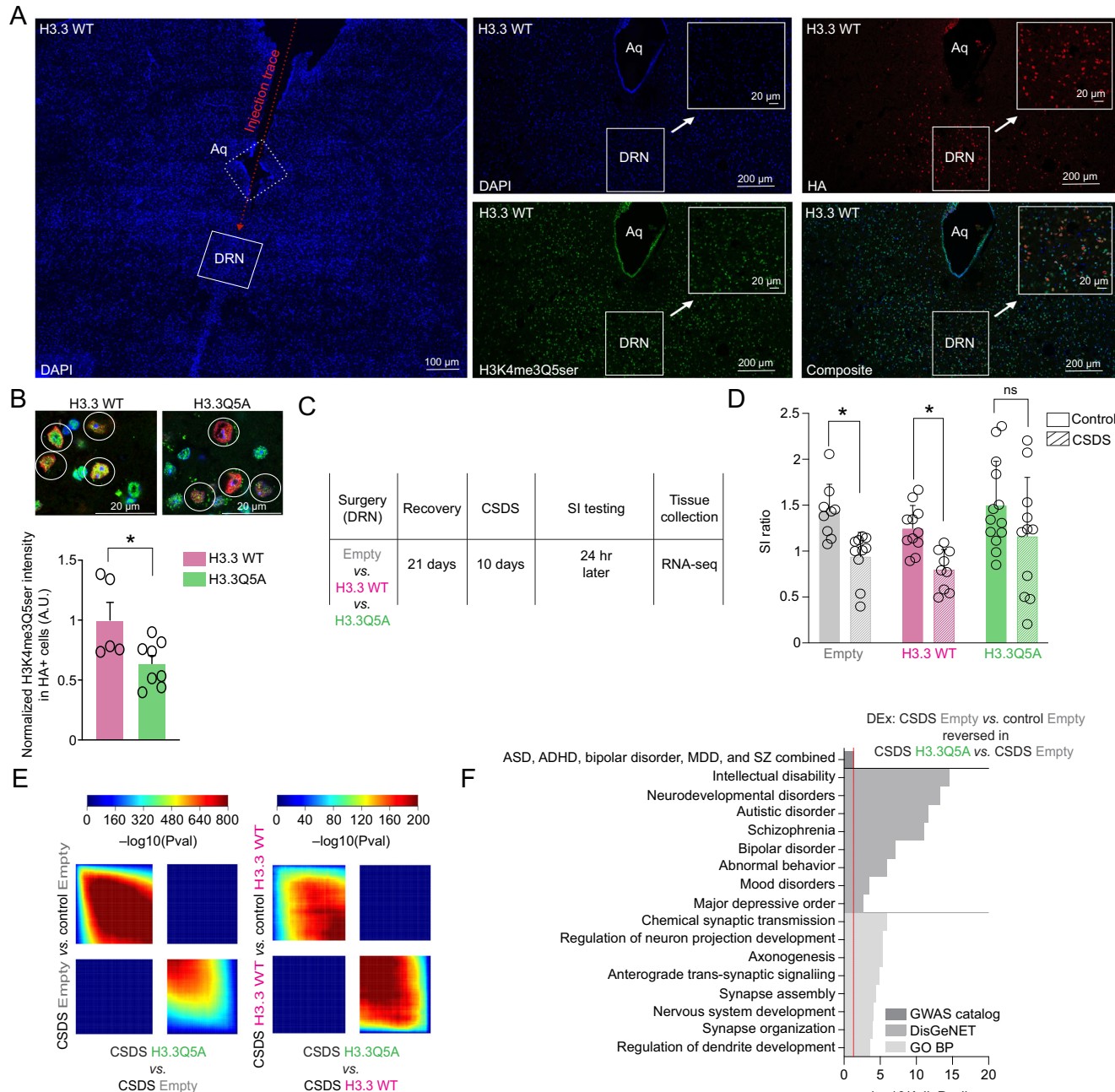

**Fig. 4 | Viral-mediated downregulation of H3 serotonylation in DRN promotes stress resilience and attenuates stress-induced gene expression. A** IHC/IF images of mouse DRN virally transduced to express HA-tagged H3.3 WT—Far left panel: tiled ×40 image of DRN-containing slice with nuclear co-stain DAPI (4′,6-diamidino-2-phenylindole) showing an example injection trace to target mouse DRN; Middle and far right panels: tiled ×40 images of DRN-containing slice stained for DAPI, HA and H3K4me3Q5ser demonstrating targeted and nuclear expression of H3.3 WT. **B** Immunofluorescence-based quantification of H3K4me3Q5ser levels/intensity in DRN tissues transduced (HA+) with either H3.3WT ($n = 6$ slices) or H3.3Q5A ($n = 7$ slices) (slices analyzed from 3 animals/virus—see Supplementary Fig. 8 for slices used in quantifications). Student's two-tailed $t$ test revealed a significant difference between H3.3Q5A vs. H3.3 WT transduced mice ($p = 0.0444$, $t_{11} = 2.269$); representative zoomed in ×40 images of quantified DRN cells are provided (co-stained for DAPI, HA and H3K4me3Q5ser). **C** Experimental timeline for male CSDS after intra-DRN viral transduction by empty vector, H3.3 WT or H3.3Q5A vectors, followed by behavioral testing and tissue collections for RNA-seq. **D** SI ratios of GFP ($n = 9$ control; $n = 10$ CSDS), H3.3 WT ($n = 11$ control; $n = 9$ CSDS) and H3.3Q5A ($n = 13$ control; $n = 11$ CSDS) transduced mice, control vs. CSDS. Two-way ANOVA significant main effects of stress observed ($p = 0.0001$, $F_{1,57} = 17.29$). Bonferroni's MC tests within viral group revealed significant differences between control vs. CSDS groups in GFP ($p = 0.0310$) and H3.3 WT mice ($p = 0.0474$), with no differences observed between control vs. CSDS H3.3Q5A mice. **E** Threshold-free RRHO analyses comparing transcriptional profiles for stress-regulated genes in empty vector and H3.3 WT-transduced DRN (control vs. CSDS) to H3.3Q5A-transduced DRN from CSDS mice ($n = 4–9$/group). Each pixel represents the overlap between differential transcriptomes, with the significance of overlap of a hypergeometric test color-coded. **F** Pathway enrichment for PCGs displaying differentially expressed genes in CSDS empty vs. control empty comparisons and rescue in CSDS H3.3Q5A vs. CSDS Empty comparisons (FDR < 0.1). Select enriched pathways are shown (FDR < 0.05; Benjamini–Hochberg). For all bar graphs, data presented as mean ± SEM. Source data are provided as a Source Data file.

(Supplementary Fig. 6E), with these overlapping genes displaying significant enrichment (FDR < 0.05) for pathways/processes (GO Biological process) related to synaptic organization/function, as well as disease associated pathways (GWAS catalog) related to mood disorders (e.g., MDD, bipolar disorder, depressive symptoms) and other psychiatric illnesses (Supplementary Fig. 6F, Supplementary Data 68, 69). In addition, we found that 37% of all PCGs exhibiting differential enrichment for H3K4me3Q5ser in human MDD cases + ADs vs. MDD subjects−ADs at their time of death overlapped with genes displaying significant differential expression between H3.3Q5A vs. empty CSDS mice, again indicating that reductions in H3K4me3Q5ser in DRN may contribute importantly to the regulation of genes associated with AD responsiveness. In sum, our viral manipulation data demonstrate that downregulation of H3K4me3Q5ser in DRN of chronically stressed mice is causally sufficient to reverse stress-mediated transcriptional programs and promote behavioral resilience. However, whether such downregulation of the mark following stress exposures (as opposed to prophylactic inhibition, as in the experiments presented above) would also be sufficient to ameliorate stress-induced deficits remains to be elucidated in future studies.

## Discussion

Here, we demonstrated that DRN, the primary hub of serotonergic projection neurons in the central nervous system, displays robust transcriptional changes as a consequence of chronic social stress in both male and female mice. The biological processes predicted to be affected by chronic stress-related gene expression programs were found to be largely overlapping between the two sexes and significantly implicated disease associations with psychiatric and/or mood-related disorders, including MDD. These alterations in gene expression coincided with disruptions in H3 serotonylation dynamics in both male and female DRN, with similar results observed in postmortem tissues from individuals diagnosed with MDD. Interestingly, male mice deemed to be stress-resilient following CSDS displayed a significant attenuation of these H3K4me3Q5ser dynamics, indicating that patterns of differential H3K4me3Q5ser enrichment observed in stress-susceptible mice may contribute importantly to maladaptive behaviors elicited by chronic stress. We also observed in animals classified as being stress-susceptible (vs. stress-resilient) that the mark displayed aberrant accumulation in DRN during protracted periods following stress exposures and was largely restored in response to chronic fluoxetine exposures, treatments that significantly reversed behavioral deficits observed in susceptible animals. Finally, we showed that directly reducing levels of H3 serotonylation in DRN prior to CSDS promoted behavioral resilience to chronic stress and significantly rescued stress-mediated gene expression programs, with many of the same genes displaying regulation by both chronic fluoxetine exposures and direct manipulations of H3Q5ser itself. In sum, these data establish a non-canonical, neurotransmission-independent role for 5-HT in the precipitation of stress-induced gene expression programs and maladaptive behavioral plasticity in DRN, results that suggest potential alternative roles for this important molecule in affect-related pathophysiology and the treatment of such disorders by classical SSRI ADs.

While the '5-HT hypothesis of depression' remains highly influential, largely owing to the fact that most currently prescribed ADs act pharmacologically to increase 5-HT signaling in brain (as well in peripheral systems), a paucity of data exists directly implicating disruptions in serotonergic signaling/neurotransmission in the precipitation of disease. In fact, one recent meta-analysis attempting to link 5-HT (as well as the 5-HT metabolite 5-HIAA) concentrations in body fluids, serotonin 5-HT$_{1A}$ receptor binding, SERT levels via imaging or at postmortem, tryptophan depletion studies or SERT gene-environment interactions to MDD pathology identified only weak, and often inconsistent evidence of interactions between these phenomena and MDD diagnosis in humans[5]. Here, we posit that additional, previously undescribed 5-HT-related mechanisms may also contribute importantly to the pathophysiology of stress/mood-related disorders and should be considered in future studies aimed at examining functions for this molecule as a precipitating factor in disease.

Given that H3 serotonylation functions independently of neurotransmission and is critically important for both the establishment and maintenance of normal gene expression programs in brain, our observation that chronic stress, widely accepted as a major contributor to MDD pathology and incidence levels in humans, significantly alters baseline patterns of H3K4me3Q5ser in DRN−a phenomenon that if rescued (either through the use of viral vectors or chronic AD treatments), appears sufficient to restore stress-mediated gene expression and promote behavioral resilience−suggests that elementary correlations between 5-HT signaling (i.e., 5-HT levels and/or receptor binding) and MDD diagnosis may be insufficient to fully elucidate roles for this molecule in affect-related disorders. Furthermore, we hypothesize that these findings may help to explain the delayed efficacy of 5-HT associated ADs in both humans with MDD and preclinical rodent models. Many of our previous findings have suggested that H3 monoaminylation levels are largely dictated by intracellular donor (i.e., monoamine) concentrations[20], but once established in neural chromatin, it remains unclear how quickly the mark will be turned over, especially given the relatively slow kinetics of histone turnover observed in both neurons and glia[50]. It is also unclear at this time precisely how histone serotonylation dynamics are mechanistically regulated in response to chronic stress. For example, it is possible that chronic stress exposures result in alterations in serotonin biosynthesis, which leads to the accumulation of intracellular donor pools of serotonin, thereby allowing for increased deposition of the mark during protracted vs. acute periods following stressful experiences. It is also possible that chronic stress induces aberrant regulation of the H3 serotonylase, TGM2, which may then alter H3 serotonylation dynamics following stress exposures. While beyond the scope of the current study, these important questions merit further investigation and may help to elucidate the transcriptional and behavioral consequences of distinct patterns of regulation observed for the mark at time points immediately following stress (when H3K4me3Q5ser is globally decreased) vs. protracted periods after CSDS (when H3K4me3Q5ser accumulates). While our transcriptional and behavioral data demonstrating that prophylactic blockade of H3 serotonylation dynamics is sufficient to increase stress-resilience and attenuate stress-induced gene expression, these data may, at first glance, appear at odds with our western blotting findings indicating that stress-susceptibility at acute time points following stress (as also observed in major depressive disorder) correspond with globally decreased levels of the mark in DRN. However, we posit that it is not necessarily the overall abundance of the mark that dictates stress-susceptibility, but rather which specific genes display stress-induced dynamics of the mark, with both increased and decreased enrichment at genomic loci effectively disrupting the homeostasis of genes implicated in affective disorders. Future studies aimed at identifying how genes and/or gene sets that display these 'aberrant' dynamics are targeted by serotonylation to precipitate the behavioral deficits observed will indeed be needed decipher the full extent of H3 serotonylation's mechanisms of action for potential future therapeutic targeting.

From a therapeutic stand point, it is our hypothesis that AD treatments may then function, at least in part, to increase 5-HT release from serotonergic neurons, thereby reducing intracellular 5-HT concentrations, eventually leading to loss, or restoration, of the mark within these cells. However, if the mark remains relatively stable during initial AD treatments, then its accumulation may not be fully resolved by acute administrations of these drugs. If true, then chronic treatments with ADs may be required to facilitate the full restoration of

normal H3 serotonylation levels in serotonergic neurons, only with time would aberrant stress-induced gene expression programs be appropriately corrected. Further investigations will be needed to fully elucidate the precise kinetics of H3 serotonylation turnover in DRN as a consequence of AD treatments in order to demonstrate whether such dynamics are indeed causally linked to symptomatic alleviation of stress-related phenotypes. In addition, it also remains unknown how altering H3K4me3Q5ser levels in important 5-HTergic projection regions, such as mPFC, might affect phenotypic outcomes resulting from chronic stress and/or AD exposures. This will be important given that chronic fluoxetine exposures might be expected to simultaneously reduce H3K4me3Q5ser levels in DRN (which ameliorates stress-induced phenotypes) while increasing the mark in regions of the brain receiving 5-HTergic innervation. Such potential phenomena will require future studies to fully be resolved. It is also worth pointing out that while direct viral-mediated alterations in H3 serotonylation dynamics in DRN of non-stressed mice do not appear to impact baseline anxiety-related measures (e.g., OFT, EPM), certain SSRI ADs, such as fluoxetine, have proven useful in clinically treating anxiety disorders and anxiety-related symptoms in individuals with MDD. Although we have yet to explore the impact of disrupting H3 serotonylation dynamics on stress-induced anxiety-related behaviors, such observations may indicate an important area of divergence between the genomic mechanisms controlled by H3 serotonylation vs. the pharmacological consequences of SSRI treatments. Given this, it will be critically important in future studies to elucidate the entirety of behavioral consequences of altering H3 serotonylation in brain (and across different brain regions implicated in mood) in order to determine whether targeting this mark—or its associated chromatin regulatory machinery—holds promise in the treatment of affect- and anxiety-related disorders. And even if the direct targeting of H3 serotonylation (or any other histone PTM for that matter) in human brain for the treatment of MDD or other affective disorders proves difficult given its critical baseline functions as a permissive chromatin modification, it may be possible to target its writer enzyme, TMG2, or other interacting proteins (e.g., potential readers, which have yet to be identified) in order to indirectly affect the mark's control over aberrant stress-induced gene transcription. While much remains to be learned regarding H3 serotonylation's precise mechanisms of actions in cells, we posit that such explorations will prove useful in helping to decipher the complex roles that serotonin plays in the regulation of mood, and how such mechanisms may be exploited in future therapeutic efforts.

Additionally, while our current study is focused primarily on alterations in H3 serotonylation dynamics in DRN as a putative precipitating factor in stress-related gene expression programs and behavior, it is important to note that DRN is not a homogeneous monoaminergic brain structure, as it has been shown previously that a smaller population of dopaminergic neurons also reside in DRN and can contribute importantly to certain affect-related behaviors[51]. This is of particular interest given that our previous work also identified dopamine as an important donor molecule for H3Q5 transamidation in brain, a modification (i.e., H3Q5dop) that we showed accumulates in ventral tegmental area (VTA) of rats during abstinence from chronic, volitional administration of cocaine and heroin[19,25]. H3Q5dop accumulation was found to potentiate aberrant gene expression programs in VTA that contribute to hyper-dopamine release dynamics in response to drug cues and increased vulnerability to drug relapse-related behaviors[19]. Like that of H3 serotonylation in DRN, which displayed acute downregulation following CSDS (24 hr after SI testing) and subsequent accumulation during protracted periods after chronic stress exposures, H3Q5dop was also found to be reduced in VTA immediately after drug administration, dynamics that were reversed during drug abstinence and were found to promote persistent maladaptive plasticity and increased cue-induced craving for drugs of abuse. Consistent with these earlier drug abuse studies, we found that the persistent accumulation of H3 serotonylation in DRN following chronic stress exposures influenced the potentiation of stress susceptibility. In addition, while at first glance, our data demonstrating that H3K4me3Q5ser levels (via western blotting) were reduced in individuals diagnosed with MDD (without ADs onboard at their time of death) may appear to contradict our mechanistic findings that H3 serotonylation accumulation in DRN is most tightly associated with stress-susceptibility, we posit that such reductions in human DRN are likely reflective of the agonal state of the subjects examined, as nearly all of the MDD −AD individuals included in this study died by suicide. Thus, it is possible that the molecular alterations in H3 serotonylation levels being captured in our data more closely resemble periods of ongoing stress, which would be consistent with our rodent data from 24 hr post-SI testing. Similar results were observed for H3Q5dop in VTA of postmortem subjects diagnosed with cocaine-dependence and who died by drug overdose, where we found that their global levels of H3Q5dop were downregulated and more closely resembled periods of active drug-taking in rodents[19]. Thus, while comparisons of such molecular phenomena in preclinical rodent models vs. clinically diagnosed humans remain grossly informative, these types of postmortem human analyses may not faithfully inform on the precise mechanistic roles for H3 serotonylation in disease etiology, thereby further highlighting the importance of using well-controlled, preclinical models for the study of complex psychiatric disorders. It is important to note, however, that AD treatments in MDD patients were observed to renormalize total levels of H3K4me3Q5ser (via western blotting) in DRN with a greater number of PCGs displaying loss of the mark (as assessed via ChIP-seq), data that are consistent with the effects of chronic fluoxetine treatments observed in stress-susceptible animals. These findings indicate that alterations in the mark's enrichment observed in behaviorally responsive, fluoxetine treated CSDS mice may indeed be of clinical relevance and may reflect functional chromatin adaptations that occur in human MDD subjects undergoing AD treatments.

An additional limitation of the current study is the possibility that our viral dominant negative approach may also impact H3Q5dop in DRN, as both marks are indeed present within this brain region (note that H3Q5his is only very weakly found within DRN[20]), although their relative stoichiometries remain unclear. Presumably, given that the proportion of serotonergic vs. dopaminergic neurons is largely skewed towards that of serotonergic cells in DRN, one might assume that the serotonylation mark would be more dominantly expressed, though this has yet to be tested empirically. While H3K4me3Q5ser and H3K4me3Q5dop are predicted to have similar molecular functions (e.g., recruiting the same reader proteins) it will be important in future studies to develop methodologies that can selectively target each modification independently (note that no such methodologies currently exist), followed by examinations of whether H3K4me3Q5dop (vs. H3K4me3Q5ser) is similarly responsive to chronic stress exposures in DRN. Further investigation of monoaminyl marks in other brain structures and cell populations beyond monoaminergic neurons may also uncover distinct regional or cell type-specific mechanisms that influence neuronal signaling and behavior. Finally, while histone H3 has been demonstrated to be a critical substrate for monoaminylation events in brain, future studies aimed at uncovering the full repertoire of monoaminylated proteins in brain, as well as their responsiveness to chronic stress exposures and AD treatments, may prove informative to the understanding of how alterations in monoaminergic activities may contribute to MDD pathophysiology and its treatment.

## Methods
### Animals
C57BL/6 J mice were purchased from The Jackson Laboratory. Retired male CD-1 breeders of at least 4 months of age were purchased from Charles River laboratories and used as aggressors. All mice were singly

housed following CSDS and maintained on a 12-h/12-h light/dark cycle throughout the entirety of the experiments. Mice were provided with *ad libitum* access to water and food throughout the entirety of the experiments. All animal procedures were done in accordance with NIH guidelines and with approval with the Institutional Animal Care and Use Committee of the Icahn School of Medicine at Mount Sinai.

## Male CSDS

Male chronic social defeat stress (CSDS) was performed, as previously described[36]. Briefly, CD-1 retired breeders were screened for aggressive behavior and were then single-housed in static hamster cages on one side of a clear perforated divider 24 hr prior to the start of CSDS. For 10 min every day, for 10 days, 8-week old C57BL/6 J experimental mice were placed in the same side of the home cage as the CD-1 mouse. The CD-1 mouse was then allowed to physically attack the intruder C57BL/6 J mouse throughout the 10-min defeat session. After each defeat session, experimental mice were moved to the opposite side of the clear perforated divider for 24 hr, permitting sensory interactions with the aggressor. Experimental mice were then rotated to a new cage with a novel aggressor every day for the remainder of the experiments. 24 hr after the final defeat, experimental mice were single-housed in static mouse cages for subsequent social interaction testing.

*Controls*: 8-week old C57BL/6 J control mice were pair-housed in mouse cages on either side of a clear perforated divider, similar to the ones used in hamster cages. Each control mouse was exposed to a novel mouse daily via rotation in a similar fashion to the experimental animal, but was never exposed to a CD-1 aggressor. Control mice were single-housed in static mouse cages at the end of the 10-d experiment for subsequent social interaction testing.

All behavioral protocols adhered strictly to the Guidelines for the Care and Use of Mammals in Neuroscience and Behavioral Research (National Academies Press, Washington, DC, 2003). All animals subjected to any form of stress were carefully monitored for their health and wellbeing in concert with the Icahn School of Medicine at Mount Sinai's veterinary staff. Any animals showing untoward effects of stress were euthanized. In our experience, such untoward effects are extremely rare (<3% of all animals studied).

## Male SI testing

24 hr after completion of CSDS, mice were tested for social avoidance via social interaction testing, as described previously[36]. Briefly, in this test, animals were transferred to a quiet room under red-light conditions and were habituated for 30 min to 1 hr prior to testing. For the first session, the subject animal was placed in a novel open-field arena with a small, wired enclosure on one side of the arena. The mouse was allowed 2.5 min to explore the empty arena, and its baseline exploration behavior was tracked from above via a video camera connected to a computer running Ethovision tracking software. In the second session, a novel CD-1 mouse was placed in the small enclosure in the arena, and the subject mouse was placed back in the arena for another 2.5 min, and exploration behavior was tracked via EthoVision. Social interaction was assessed by SI ratio, which is the amount of time the animal spent in the interaction zone while the CD-1 mouse was present, over the time spent in the interaction zone while the CD-1 was absent. A subject mouse was deemed to be stress-resilient if it had an SI ratio greater than 1, whereas stress-susceptible mouse had SI ratios less than 1.

## Female CSDS

Female social defeat was performed as previously described[47]. Briefly, intact female Swiss Webster (CFW) mice were housed with castrated male mice and were tested for aggression against experimental female intruder mice. Wild-type 12-week old female C57BL/6 J (B6) mice were socially defeated daily by aggressive CFW female resident mice for 5 min per day during the 10-day paradigm. Between the defeats, experimental B6 female mice were housed with the aggressor female in a shared home cage, separated by a clear perforated cage divider. Control females were housed in identical conditions but were never exposed to a physical defeat. Defeated and control females were singe housed following the final defeat.

## Female SI testing

Social interaction testing was done in the experimental female's home cage 24 h after the final defeat. In this test, a non-aggressive B6 female was placed into the experimental female's home cage for 1.5 min and social interaction time and defensive score was assessed. Social interaction included any anogenital, flank, naso-nasal sniffing, or flank on flank contact that was initiated by the experimental animal. Defensive score was defined on a numerical scale from 0 to 3, with 0 being not defensive, 1 being minimally defensive (avoidance only), 2 being moderately defensive (avoidance, digging, but no kicking), and 3 being highly defensive (avoidance, escape, kicking, flinching, digging, jumping, pacing). Tissue was collected 24 h after the social interaction test (i.e., 48 h after the final defeat). As in previous reports using this female CSDS paradigm[47], vaginal cytology was monitored in experimental mice during the 10-day social defeat protocol using the lavage technique. Consistent with the literature, CSDS did not affect estrous cycling (nor body weight) in defeated females. Since we did not have any evidence to suggest that estrous stage significantly impacts female responsiveness to CSDS, we did not use it as a covariate in our sequencing analyses.

## Fluoxetine treatments

24 hr following social interaction testing, each group of male mice (control, stress-susceptible and stress-resilient) were randomly separated into two groups, either to receive regular drinking water (vehicle) or drinking water with fluoxetine hydrochloride for 30 d. Drug treatment was performed as previously described[49]. Briefly, fluoxetine hydrochloride (Spectrum Chemical) was administered *ad libitum* in drinking water (filtered tap water) in opaque light-protected bottles (Argos Technologies Litesafe Centrifuge Tubes 50mL- Fisher, #03-395-120). Fluoxetine solutions were changed and refreshed 3 times per week. Fluoxetine was administered through drinking water at 160 mg/L. Water was weighed every day to monitor consumption and track dosage. Mice drank ~2–3 ml per day of 160 mg/L solution, resulting in an estimated 15.25 mg/kg dose over the treatment period. Following completion of 30 d of treatment, mice underwent social interaction testing to evaluate drug efficacy.

## Forced swim test (FST)

The forced swim test was conducted as previously described[52]. Briefly, mice were placed in a 4-liter glass beaker with 2 L of room-temperature water for 6 min. Each session was recorded and hand-scored, recording the number of seconds the mouse was immobile.

## Open field test (OFT)

Open field testing was performed as previously described[52]. Briefly, mice were placed in a 16 ×16-inch open field apparatus under dim lighting and distance and time in center vs. periphery were recorded via Ethovision software.

## Elevated plus maze (EPM)

The elevated plus maze was used as previously described[52]. Briefly, mice were placed into the center of the maze under dim lighting and allowed to explore for 5 min. Time spent in the closed and open arms and number of explorations of open arms was recorded via Ethovision software, as previously described[52].

## Human brain samples

Human DRN tissues from the Dallas Brain Collection (UT Neuropsychiatry Research Program) were obtained from the Southwestern Institute of Forensic Sciences at Dallas, UT Southwestern Transplant Services Center, and UT Southwestern Willed Body Program, following consent from donor subjects' next of kin, permission to access medical records and to hold direct telephone interviews with a primary caregivers. All clinical information obtained for each donor was reviewed by three research psychiatrists, using DSM-V criteria for diagnoses. Blood toxicology screens were conducted for each donor subject from the Southwestern Institute of Forensic Sciences at Dallas. Collection of postmortem human brain tissues is approved by the University of Texas Southwestern Medical Center Institutional Review Board [STU 102010-053]. Brain tissue dissections were removed, frozen immediately using dry ice and 2-methylbutane (1:1, v:v) and stored at −80 °C. For western blotting validation experiments, H3 was used an internal reference control for the best normalization and most reliable indicator of equal protein concentration. Demographic information can be found in Supplementary Data 70.

## RNA isolation, RNA-seq and analysis

For male and female CSDS experiments, DRN tissues were collected from mice (following rapid decapitation) 24 hr after final social interaction (1 mm punches) and immediately flash-frozen. To examine genome-wide effects of blocking serotonylation via viral infection, brains were sectioned at 100 μm on a cryostat, and GFP/RFP was illuminated with a NIGHTSEA BlueStar flashlight to microdissect virally infected tissues. DRN tissue punches were homogenized in Trizol (Thermo Fisher), and RNA was isolated on RNeasy Microcolumns (Qiagen) following manufacturer's instructions. Following RNA purification, RNA-seq libraries were prepared according to the Illumina Truseq RNA Library Prep Kit V2 (#RS-122-2001) protocol and sequenced on the Illumina Novaseq platform. Following sequencing, data was pre-processed and analyzed as previously described[19]. Briefly, FastQC (Version 0.72) was performed on the concatenated replicate raw sequencing paired-end reads from each library to ensure minimal PCR duplication and sequencing quality. Reads were aligned to the mouse mm10 genome using HISAT2 (Version 2.1.0) and annotated against Ensembl v90. After removal of multiple-aligned reads, remaining reads were counted using featurecounts (Version 2.0.1) with default parameters, and filtered to remove genes with low counts (<10 reads across samples). For male 24-hr post-CSDS RNA-seq, RUVr[53], k = 6, was performed to normalize read counts based on the residuals from a first-pass GLM regression of the unnormalized counts on the covariates of interest. For female RNA-seq experiments and the serotonylation manipulation experiments with the Q5A virus, RUVr[53] (female; k = 4, Q5A; k = 6 was performed to normalized read counts. DESEQ2[54] (Version 2.11.40.6) was used to perform pairwise differential expression analyses between indicated comparisons. Differentially expressed (DE) genes were defined at FDR < 0.05. Unsupervised clustering heatmaps were generated at DE genes across samples using heatmap2 from gplots (Version 3.1.3). Threshold free Rank-Rank Hypergeometric Overlap (RRHO) maps were generated to visualize transcriptome-wide gene expression concordance patterns as previously described[55], using RRHO2 (Version 1.0). Odds ratios for overlapping gene sets were calculated with GeneOverlap (Version 1.34.0). Enrichment analysis on gene sets of interest was performed with EnrichR, Benjamini–Hochberg (BH) q values corrected for multiple testing are reported[56–58].

## Western blotting and antibodies

DRN tissues were collected (following rapid decapitation) from mice (1 mm punches) and immediately flash-frozen. Punches were homogenized using a sonicator in RIPA Buffer, containing 50 mM Tris-HCl, 150 mM NaCl, 0.1% SDS, 1% NP-40 and 1× protease inhibitor cocktail. Protein concentrations were measured using the DC protein assay kit (BioRad), and 20 ug of protein was loaded onto 4-12% NuPage BisTris gels (Invitrogen) for electrophoresis. Proteins were then fast-transferred using nitrocellulose membranes and blocked for 1 hr in 0.1% Tween-20 in 1× PBS (PBS-T) in a 5% milk buffer, before undergoing overnight incubation with primary antibodies at 4 °C. The following day, blots were washed of primary antibody for 10 min 3× in PBS-T, then incubated for 1 hr with horseradish peroxidase conjugated anti-rabbit (BioRad 170-6515, lot #: 64033820) or anti-mouse (GE Healthcare UK Limited NA931V, lot #: 9814763) secondary antibodies (1:10000; 1:50000 for anti-H3 antibody, BioRad) in 0.1% Tween-20 in 1× PBS (PBS-T) in a 5% milk buffer at RT. Blots were then washed of secondary antibody for 10 min 3× in PBS-T and bands were detected using enhanced chemiluminescence (ECL; Millipore). Densitometry was used to quantify protein bands via Image J Software and proteins were normalized to total H3 or GAPDH, as indicated. For cultured cerebellar granule neuron (cGN) western blotting experiments, 1 h after 50 mM KCl treatment, cGNs in 6-well plates were rinsed with 1× PBS and lysed in 200 μl of 2× SDS loading buffer (100 mM Tris-HCl pH 6.8, 20% glycerol, 4% SDS, 0.1% bromophenol blue and 2% 2-mercaptoethanol). 15 μl of samples were loaded on 4–12% NuPAGE gel and transferred to nitrocellulose membranes. The following antibodies were used: rabbit anti-H3K4me3Q5ser (1:500, ABE2580; MilliporeSigma), rabbit anti-H3Q5ser (1:500, MilliporeSigma; ABE1791), rabbit anti-H3 (1:50000, Abcam ab1791), H4 (1:10000, Abcam; ab10158), H3.3 (1:2000, MilliporeSigma; 09-838,), FLAG (1:5000, Sigma; F3165,) and rabbit anti-Gapdh (1:10000, Abcam; ab9485).

## Chromatin immunoprecipitation

DRN tissues were collected (following rapid decapitation) from mice (1 mm punches) and immediately flash-frozen. Punches were cross-linked with 1% formaldehyde and rotated gently at room temperature for 12 min. Punches were then immediately quenched with glycine and rotated gently at room temperature for 5 min. Samples were washed thoroughly before lysis and sonications were performed, as previously described[18]. Samples were then incubated with specific antibodies (7.5 μg per sample) bound to M-280 Dynabeads on a rotator at 4 °C overnight. The following day, immunoprecipitates were washed, eluted and reverse-crosslinked. Samples underwent RNA and protein digestion and DNA was purified using a Qiagen PCR purification kit. The following antibodies were used: rabbit anti-H3K4me3Q5ser (1:500, ABE2580; MilliporeSigma).

## ChIP-seq library preparation and analysis

Following DNA purifications, ChIP-seq libraries were generated according to Illumina protocols and sequenced on an Illumina HiSeq2500, 4000 or Novaseq Sequencers. ChIP-seq peaks were called and differential analysis conducted exactly as described previously[18,59]. Briefly, raw sequencing reads were aligned to the mouse or human genome (mm10 or hg38, respectively) using default settings of HISAT2. Alignments were filtered to only include uniquely mapped reads using SAMtools v.1.8. Peak-calling was normalized to respective inputs for each sample and was performed using MACS v.2.1.1[60] with default settings and filtered for FDR < 0.05. Differential analysis was performed using diffReps[61] with a 1 kb window size. Peaks and differential sites were further annotated to nearby genes or intergenic regions using the region analysis tool from the diffReps package. To be considered a real peak-containing PCG, a significant peak (FDR < 0.05, >5-fold enrichment over input) had to be found in a PCG (promoter and/or gene body) in at least one of: 3 conditions for male 24 hr post-SI testing (Control, Susceptible or Resilient); 2 conditions for female 24 hr post-SI testing (Control, Defeat); 4 conditions for fluoxetine

experiments 30d post-SI testing (control −/+ FLX, SUS −/+ FLX); or 4 conditions for human DRN (MDD−ADs, MDD + ADs vs. matched controls). To be considered a differentially enriched gene, it had to first pass the aforementioned criteria, and then display a ≥1.5 or ≤ −1.5 or ≥1.0 or ≤ −1.0 fold difference between conditions (pairwise comparisons) at FDR < 0.05 (as indicated throughout). Enrichment analysis on gene sets of interest was performed with EnrichR, Benjamini−Hochberg (BH) q-values corrected for multiple testing are reported[56–58].

### ChIP/Re-ChIP experiments in cultured granule neurons

**Cerebellar granule neuron culture.** Granule neurons were prepared from cerebellum of P7 CD-1 mouse pups as previously described[62]. Briefly, on day 1 in vitro (*DIV 1*), granule neurons were transduced with AAV-empty, AAV-H3.3-WT or AAV-H3.3Q5A respectively. 2 days after infection, the medium was changed to low KCl medium (Basal Medium Eagle, GIBCO + 5% Hyclone bovine growth serum, Cytiva+1× penicillin-streptomycin, GIBCO+1× GlutaMAX™ Supplement, GIBCO + 5 mM KCl).

**ChIP and Re-ChIP-qPCR.** ChIP assays were performed with cultured granule neurons, as described previously with modifications[63]. After quenching and sonication, 15 million granule neurons and 15 µl anti-FLAG beads (Sigma, #A2220) were used for each ChIP reaction. After IP, chromatin was eluted twice with 100 µl of 3× FLAG Peptide solution (Sigma, #F4799, dissolved in ChIP lysis buffer) for 30 min at 4 °C. The two eluents were mixed and incubated with 1 µg of anti-H3K4me3Q5ser antibody (Millipore, #ABE2580) for 4h-overnight. Next, the chromatin-antibody mixture was incubated with 25 µl washed Dynabeads Protein A (Invitrogen, #10001D). The following steps were the same as the ChIP assays described above. ChIP and Re-ChIP DNA was purified using a Qiagen PCR purification kit and eluted in 60 µl elution buffer. 2 µl of ChIP or Re-ChIP DNA was used for each qPCR reaction. See Supplementary Data 71 for mouse ChIP-qPCR primers.

### Viral constructs

Lentiviral constructs were generated as previously described[18]. Briefly, lenti-H3.3 constructs [wildtype (WT) vs. (Q5A)-Flag-HA] were cloned into a pCDH-RFP vector via PCR and enzyme restriction digestion. Plasmids were purified and sent to GENEWIZ for sequence validation. pCDH-GFP-H3.3 plasmids were then sent to Cyagen Biosciences for lentiviral packaging. For cultured cerebellar granule neuron experiments, pAAV-CMV-H3.3-IRES-GFP constructs [wildtype (WT) vs. (Q5A)-Flag-HA vs. empty] were packaged as follows: 70–80% confluent HEK293T cells were transfected pAAV2/1 (Addgene 112862), pAdDeltaF6 (Addgene 112867), and pAAV-CMV-IRES-GFP or pAAV-CMV-H3.3-WT-IRES-GFP or pAAV-CMV-H3.3-Q5A-IRES-GFP with PEI reagent (Polysciences, #26008-5). 48–72 h after transfection, the media with AAVs were collected by centrifuge. AAV particles were precipitated by adding 10% volume of PEG 8000-NaCl solution (40% PEG 8000, 2.5 M NaCl, pH 7.4). Next, the AAV particles were resuspended in granule neurons culture medium. For in vivo validation experiments, where applicable, plasmids were sent to Cyagen Biosciences for high-titer packaging.

### Viral transduction

Mice were anesthetized with a ketamine/xylazine solution (10/1 mg/kg) i.p. and positioned in a stereotaxic frame (Kopf instruments). 1 µl of viral construct was infused intra-DRN using the following coordinates; anterior-posterior (AP) −4.40 mm, medial-lateral (ML) 0.0 mm, dorsal-ventral (DV) −3.40 mm. Following surgery, mice received meloxicam (1 mg/kg) s.c. and topical antibiotic treatments for 3 days. Viral validations, chronic social defeat stress and other behaviors were performed at least 21 days post surgery to allow for optimal viral expression and recovery.

### Immunohistochemistry

Mice were anesthetized with isoflurane and perfused with cold 1× PBS and 4% PFA. Brains were then post-fixed in 4% PFA overnight and then transferred into a solution of 30% sucrose/PBS 1× for two days. Following one wash with 1× PBS, brains were mounted in Tissue-Tek® O.C.T. Compound (Sakura, #4583) and sectioned on a cryostat (Leica CM3050-S) at a thickness of 40 µm, collecting serial sections from DRN. Tissue sections were stored at 4 °C with 1%PBS and 0.01% sodium azide until processing for immunofluorescence. 2–3 brain slices for each subject were washed 3× in 1× PBS for 10 min each at RT. Then, DRN slices were incubated for 30 min in 0.2% Triton X/PBS 1x, followed by a 1 hr incubation at RT in blocking buffer (0.3% Triton X, 3% normal donkey serum, 1× PBS). Finally, slices were incubated overnight at 4 °C with primary antibodies [rabbit anti-H3K4me3Q5Ser (1:500, MilliporeSigma; ABE1791), mouse anti-HA (1:1000, SantaCruz, Cat#, sc-7392)]. After three consecutive washes in 1× PBS, slices were incubated for 2 h at RT on a slow shaker with secondary fluorescent conjugated antibodies (donkey anti-rabbit AlexaFluor568−ThermoFisher A-10042−and donkey anti-mouse AlexaFluor680−ThermoFisher A-21109; 1:1000) in blocking solution. Slices were then washed 3× times for 10 mins in 1× PBS and incubated with DAPI (1:10000, Thermo Scientific 62248) for 5 min. Subsequently, slices were further washed in 1× PBS and mounted on charged thermofrost slides using ProLong Gold Antifade Mountant (Thermofisher, Cat. No. P36934).

### Immunofluorescence image analysis

Digital images were acquired using a confocal microscope (Zeiss LSM 780, upright) using the Zen Black software. A ×40 magnification with oil objective was used to acquire images from 1 to 3 DRN replicates per subject. The image acquisition was performed with a 1024×1024 frame size. Images were averaged across 8 consequent acquisitions at a bit depth of 16 bits. The wavelengths selected were 405 (DAPI), 568 (H3K4me3Q5Ser) and 680 (HA). All slides used for quantification were imaged during the same confocal session and under the same parameters, keeping the master gain, digital offset, digital gain and laser power for each wavelength consistent through images. Files were saved in.tiff format and processed via ImageJ/FiJI software. H3K4me3Q5ser intensity in transduced cells (H3K4me3Q5ser+/HA +/DAPI+) was determined using the Create Selection and Measure functions in ImageJ/FiJi. Background MFI was subtracted for each image from the MFI of the ROI, and the data are presented as normalized intensity values.

### Statistics

Statistical analyses were performed using Prism GraphPad software. For all behavioral testing and biochemical experiments involving more than two conditions, two-way or one-way ANOVAs were performed with subsequent *post hoc* analyses. For experiments comparing only two conditions, two-tailed Student's t tests were performed. Sequencing-based statistical analyses are described above. In biochemical and RNA-seq analyses, all animals used were included as separate *n*s (i.e., samples were not pooled). In ChIP-seq analyses, animals were pooled per *n* as designated above. Significance was determined at p ≤ 0.05. Where applicable, outliers were determined using Grubb's test (alpha = 0.05; noted in Supplementary Fig. 7). All bar/dot plot data are represented as mean ± SEM.

### Reporting summary

Further information on research design is available in the Nature Portfolio Reporting Summary linked to this article.

## Data availability

The RNA-seq and ChIP-seq data generated in this study have been deposited in the National Center for Biotechnology Information Gene Expression Omnibus (GEO) database under accession number

GSE216104. We declare that the data supporting the findings of this study are available within the article and Supplementary Information. No restrictions on data availability apply. Source data are provided in this paper.

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

## Acknowledgements

We would like to thank members of the Maze and Russo laboratories for critical readings of the manuscript. This work was partially supported by grants from the National Institutes of Health: R01 MH116900 (I.M.), F99 NS125774 (S.L.F.), F31 MH116588 (S.L.F.), K99 MH120334 (L.A.F.), F32 MH125634 (E.L.N.), F32 MH126534 (J.C.C.), as well as funds from MQ (I.M.) Alfred P. Sloan Foundation (I.M.), One Mind (I.M.) and Howard Hughes Medical Institute (I.M.).

## Author contributions
A.A. and I.M. conceived of the project, designed the experiments, and interpreted the data; A.A., G.D.S., S.L.F., J.C.C., L.A.F., A.E.L., R.M.B., L.K., F.C., E.L.N., C.M., P.S., Y.L., H.E.C. and S.J.R. collected and/or analyzed the data; A.A., S.L.F., A.R., L.S. and I.M. performed the bioinformatics analyses; K.G. and C.A.T. provided human postmortem tissues; A.A., G.D.S. and I.M. wrote the manuscript.

## Competing interests
The authors declare no competing interests.

## Inclusion and ethics statement
All collaborators associated with this work have fulfilled the criteria for authorship required by Nature Portfolio journals. To obtain authorship, their participation in this study was deemed to be essential for the design and implementation of the work presented. Roles and responsibilities were agreed upon among collaborators ahead of or during the research.

## Additional information

**Amni Al-Kachak**[1,8], **Giuseppina Di Salvo** [1,2,8], **Sasha L. Fulton**[1], **Jennifer C Chan**[1], **Lorna A. Farrelly**[1], **Ashley E. Lepack**[1], **Ryan M. Bastle**[1], **Lingchun Kong**[1], **Flurin Cathomas** [1], **Emily L. Newman** [3], **Caroline Menard** [1], **Aarthi Ramakrishnan**[1], **Polina Safovich**[1], **Yang Lyu**[1], **Herbert E. Covington III**[4], **Li Shen** [1], **Kelly Gleason**[5], **Carol A. Tamminga** [5], **Scott J. Russo** [1] **& Ian Maze** [1,6,7] ✉

[1]Nash Family Department of Neuroscience, Friedman Brain Institute, Icahn School of Medicine at Mount Sinai, New York, NY 10029, USA. [2]Department of Psychiatry and Neuropsychology, School for Mental Health and Neuroscience (MHeNs), Maastricht University, Maastricht, The Netherlands. [3]Department of Psychiatry, McLean Hospital and Harvard Medical School, Belmont, MA 02478, USA. [4]Department of Psychology, Empire State College, State University of New York, Saratoga Springs, NY 12866, USA. [5]Department of Psychiatry, University of Texas Southwestern Medical School, Dallas, TX 75390, USA. [6]Department of Pharmacological Sciences, Icahn School of Medicine at Mount Sinai, New York, NY 10029, USA. [7]Howard Hughes Medical Institute, Icahn School of Medicine at Mount Sinai, New York, NY 10029, USA. [8]These authors contributed equally: Amni Al-Kachak, Giuseppina Di Salvo. ✉e-mail: ian.maze@mssm.edu

