## [Peer Review File · Nature Communications]

Histone serotonylation in dorsal raphe nucleus contributes to stress- and antidepressant-mediated gene expression and behaviorReviewers' Comments:

Reviewer #1:

Remarks to the Author:

The present study provides evidence for a novel epigenetic regulatory mark, H3K4meQ5ser in the DRN, in regulating susceptibility to chronic stress and response to SSRI antidepressant treatment. Despite some limitations, the work is thorough and identifies an exciting new mechanism that may have relevance for human disease.

Major comments:

1. A major caveat to the study is the lack of validation that the dominant negative construct (Fig 4) impacts H3K4meQ5 ser levels. Thus, some statements in the results, namely in line 704 where the authors use the phrase "we found that reducing H3 705 serotonylation in DRN using the dominant negative H3.3Q5A virus..." is inaccurate. It's possible that using cultured cells from cerebellum will reflect the effects observed in vivo in DRN. The authors must address this concern to demonstrate some impact of the construct. Otherwise, it is not clear how the DN construct is impacting behavior or gene expression.
2. In Figure 2A, the statistical explanation is not very convincing for justifying a "nominal change" in the mark between control and susceptible males. In fact, the effect size is similar as observed in females (2b). Likely this experiment is simply underpowered to observe the effect via ANOVA. The authors should simply report the ANOVA results or increase the N for the control group.
3. Similarly, the human data presented in Fig 2C could be more convincing with a higher n, the data shows a very large variability in both control groups.
4. The discussion would benefit from more speculation about why the H3 mark goes in opposite directions in resilient mice and susceptible mice between "Day 11" and "Day 40" (data shown in Figs 2/3) post stress and how that may relate to serotonin levels in the DRN.

Minor comments:

1. In line 603, the authors describe 10 days of CSDS as acute and it's a little misleading/confusing.
2. It's not entirely clear why only control group data are shown for FST and EPM (Supp 5), was this a separate cohort of mice that were not defeated?
3. In line 775, the authors refer to levels of H3 ser being rescued in resilient mice...but is this really a rescue? Is it just that it never went down? I know there are other data suggesting that some CSDS induced changes initially go the same direction in Sus and Res groups then rebound, but this isn't really shown here. Maybe a little more discussion could benefit refinement of this interpretation.
4. It's a bit frustrating that so much data is in the supplements as opposed to main figures, but I appreciate the authors including the thorough characterization of their sequencing data.

Reviewer #2:

Remarks to the Author:

The manuscript "Histone serotonylation in dorsal raphe nucleus contributes to stress- and antidepressant-mediated gene expression and behavior" uses a combination of RNA sequencing, chromatin immunoprecipitation, mouse behavior, and elegant viral vector tools to demonstrate that stress susceptibility in mice and depression in humans is associated with reduced overall histone serotonylation in the DRN. Surprisingly, they also found that virally decreasing global histone serotonylation in the DRN promotes stress resilience and reverses stress-induced gene expression in mice. Using ChIP sequencing, they demonstrate specific patterns of stress-induced histone serotonylation of genes that are absent in resilient animals and reversed by chronic treatment with the antidepressant fluoxetine. Thus, it seems likely that rather than behavioral effects being driven by global levels of histone serotonylation, they are driven by altered serotonylation levels at histones associated with a specific subset of genes, and that this regulation is prevented by the viral downregulation of histone serotonylation. The authors present a great deal of RNAseq and ChIPseq

data that the field can now use to potentially determine what those specific gene targets may be. Thus, this study presents strong evidence for a novel and paradigm-shifting mechanism underlying depressive disorders and the biological functions explaining the effects of current antidepressant treatments.

Overall, this is an extremely well-written paper presenting a great deal of impactful data. The methods and statistical analyses are appropriate, the figures are clear and present the data in a useful manner, the supplemental data are useful, and the results are interpreted suitably. The introduction and discussion sections set up and interpret the experiments with copious use of the literature in a manner that is clear and readable. I have only a few minor concerns. If these issues are addressed with appropriate text edits, this manuscript will be very suitable for publication and will make an impactful addition to the field that is certain to be cited extensively.

Minor Concerns:

1) The surprising result that stress susceptibility in mice and depression in humans is associated with reduced overall histone serotonylation in the DRN and yet virally decreasing global histone serotonylation in the DRN promotes stress resilience should be addressed more directly in the discussion. Readers unfamiliar with the complexity of gene-behavior interactions may see this as a contradiction, and so the authors should spend some time at the beginning of the discussion section explaining how these results are actually consistent with their interpretations.

2) The data in supplemental figure 5 are quite interesting. They show that reducing histone serotonylation in DRN does not prevent stress-induced anxiety-related behaviors. However, fluoxetine is effective in treating anxiety disorders and anxiety-related symptoms in depression patients. This begs the question of whether histone serotonylation in other brain regions (amygdala?) could contribute to these behaviors, or whether SSRIs work via different mechanisms to affect mood vs anxiety. Experiments to investigate this are beyond the scope of the current study, but some speculation in discussion is warranted.

3) The authors should speculate in the discussion on how their novel findings could be leveraged to improve current treatments or drive new treatments for mood disorders. This includes proposing future experiments to uncover the enzymatic mechanisms regulating histone serotonylation in the brain and how these could be pharmacologically manipulated.

Reviewer #3:

Remarks to the Author:

Al-Kachak et al. characterized the impact of stress/depression and antidepressant treatment on the status of recently-discovered histone serotonylation (H3K4me3Q5ser), associated with transcriptional permissiveness. The authors found that chronic social stress alters H3K4me3Q5ser in the dorsal raphe among both male and female mice, which is rescued by antidepressant treatment. Of particular translational relevance, postmortem human dorsal raphe was also depleted of H3K4me3Q5ser among MDD subjects, but restored with concurrent antidepressant usage. However, in mice, preventing serotonylation by expressing a dominant-negative Q5A promotes resilience. This is an interesting and important study, addresses SABV, has taken advantage of cutting-edge techniques and analyses, and is extremely well-written. I have mostly minor comments/questions.

Main comments:

1. The original findings shown in Fig2 were that susceptibility/MDD reduced H3K4me3Q5ser, and antidepressants restored levels. However, results presented in Fig3 show the opposite, with susceptibility increasing H3K4me3Q5ser and FLX reducing it. The authors are careful to say "dysregulated/disrupted," and write that this difference is likely due to acute measurements of

H3K4me3Q5ser after CSDS originally, vs long-term measurements one month later with H2O/Flx treatment. However, the time scale differences do not explain the directions observed from human brain, where depression and AD treatment are also at longer time scales. The authors speculate in the discussion about intracellular stores and histone turnover rates (again related to time scales), or perhaps brain region specificity (although all conflicting measurements were in DRN), or cause of death. But it seems like some simpler experiments could begin to address this. Does acute SSRI treatment increase Q5ser? Do MAOI's (with distinct monoamine-targeting mechanism) increase or decrease Q5ser with acute vs chronic treatment? The paper deserves to be published even without these experiments or if this is not resolved. It's a great Discussion, truly – the authors dive right in and even use this as a call for further well-controlled animals studies and I almost feel bad harping on this point more here.

Minor comments:

1. Have other studies found that 5HT itself is reduced among susceptible mice in DRN? Is the (initial) decrease in 5HT binding to H3Q5 a matter of less or depleted 5HT?
2. Figure 1 A vs D: Why use different behavioral scoring methods for males vs females?
3. There's a lot going on with the colors. I know it's a painful request, but any chance the control males and females could be the same color, and the defeated females could also be blue like susceptible males? That way at least the green/purple scheme used to denote males vs females doesn't overlap with the control/defeat female color scheme (it's confusing to know whether we are looking at sex differences or CvsD females in the venn diagrams in 2, for example, and the Sus circle in H should at least match the Sus male color). Is it also possible to make the Fig 3 sus/res colors match the previous sus/res colors (with the different types of hashing for Post-H2O vs Post-FLX, that part works).
 - a. Are the groups in Fig3D all pre-treatment, based on lack of hashing? Wouldn't they have to be post-treatment? If so, this doesn't appear to match the patterns from 3B-C. I get the use of new colors for 3E-F (and another one in 3H), but... it would be much easier for the reader to connect all of the dots if color was used consistently with the previous parts of the figure. Shade instead of texture could denote treatment.
 - b. Other main and supplemental figures would also benefit from consistency.
4. WRT behavior in Fig4C: The lack of a previous statistical difference with H3.3Q5A (particularly given a visual difference between the control and CSDS groups) is not itself significant. Please report the associated p-value and effect size in the legend alongside the other values. Is there a difference between CSDS WT and CSDS Q5A? Even if these numbers are less convincing, the RRHO's are still very convincing.
5. Supp Fig2: Why are only the control genomic compartment %s shown? Did stress/susceptibility alter these distributions at all?
6. Supp3: Recommend increasing the RRHO scale.

Response to Referees

Reviewer #1:

The present study provides evidence for a novel epigenetic regulatory mark, H3K4meQ5ser in the DRN, in regulating susceptibility to chronic stress and response to SSRI antidepressant treatment. Despite some limitations, the work is thorough and identifies an exciting new mechanism that may have relevance for human disease.

Response: We very much thank the Reviewer for their positive and constructive comments on our manuscript, and their feeling that this work is “thorough and identifies an exciting new mechanism that may have relevance for human disease.” As discussed below, we have now added new data and discussions to the Resubmission to directly address the Referee’s previous concerns. Please find our detailed responses to the comments raised below.

Major comments:

1. A major caveat to the study is the lack of validation that the dominant negative construct (Fig 4) impacts H3K4meQ5ser levels. Thus, some statements in the results, namely in line 704 where the authors use the phrase “we found that reducing H3 serotonylation in DRN using the dominant negative H3.3Q5A virus...” is inaccurate. It's possible that using cultured cells from cerebellum will reflect the effects observed in vivo in DRN. The authors must address this concern to demonstrate some impact of the construct. Otherwise, it is not clear how the DN construct is impacting behavior or gene expression.

Response: We thank the Reviewer for bringing up this important point. As indicated by the Referee, in our first submission, we provided data in cultured cerebellar granule neurons (cGNs) demonstrating that incorporation of dominant negative histone H3.3Q5A into neuronal chromatin is sufficient to reduce H3K4me3Q5ser enrichment at target loci (e.g., immediate early gene promoters; **Fig. S4A-D**). While we had validated the efficacy of this viral approach *in vivo* in previous publications in terms of our ability to reduce global H3 monoaminylation states and alter associated gene expression patterns and behavior (Farrelly et al., *Nature*, 2019; Lepack et al., *Science*, 2020; Fulton et al., *Neuropsychopharmacology*, 2022; Stewart et al., *Mol Cell Neuroscience*, 2023; Sardar et al., *Science*, 2023), we have not yet published evidence definitively demonstrating that chromatin incorporation of H3.3Q5A is sufficient to reduce the mark’s enrichment at target gene loci in neural cells. Thus, this cGN validation system was chosen for two distinct reasons: 1) cGNs are cultured in serum (unlike many other neuronal culture models, such as primary cortical neurons), so they have a serotonin source that allows for the establishment of endogenous H3K4me3Q5ser; and 2) we needed to use a system that allowed for a sufficient amount of cells to provide enough chromatin input material for ChIP-/Re-ChIP experiments (ChIP’ing first for the tagged H3.3 construct in chromatin, followed by Re-ChIPs for H3K4me3Q5ser). While we feel that these experiments provided much needed evidence that incorporation of H3.3Q5A into neuronal chromatin results in loss of the mark (data that we have chosen to keep in the manuscript), we understand that these cells are distinct from those found in DRN *in vivo*. As such, we now provide additional immunohistochemistry evidence demonstrating that H3.3Q5A (vs. H3.3 WT) transduction in DRN similarly results in significant

global reductions of H3K4me3Q5ser levels in transduced cells (new **Fig. 4A-B**). While the effect *in vivo* is somewhat modest (~48% reduction of the mark in transduced cells), we believe this result is consistent with our *in cellulo* results, as well as known rates of histone turnover in neural cells (Maze et al., *Neuron*, 2015). While it is unfortunately not feasible to perform ChIP-/Re-ChIP experiments using virally infected DRN tissues *in vivo* owing to a lack of suitable input material, we feel that these new data, combined with the data presented from cGNs, nicely demonstrate the efficacy of our viral approach.

2. *In Figure 2A, the statistical explanation is not very convincing for justifying a "nominal change" in the mark between control and susceptible males. In fact, the effect size is similar as observed in females (2b). Likely this experiment is simply underpowered to observe the effect via ANOVA. The authors should simply report the ANOVA results or increase the N for the control group.*

Response: We apologize for any confusion with respect to the statistical tests used in this experiment. We now report the ANOVA results only, as suggested by the Referee, which indicate that CSDS in male mice results in a significant difference in H3K4me3Q5ser levels in DRN when comparing stress-susceptible vs. stress-resilient animals. The text has now been updated to more accurately reflect the statistical effects observed.

3. Similarly, the human data presented in Fig 2C could be more convincing with a higher n, the data shows a very large variability in both control groups.

Response: While we agree that including a higher *n* would be ideal for the western blotting data presented in **Fig. 2C**, we are unfortunately limited by human postmortem DRN tissue availability from our collaborators at the Dallas Brain Bank (DBB; Dr. Carol Tamminga and Ms. Kelly Gleason). In order to obtain additional paired tissues for this purpose would require either prospective collections from the DBB or establishment of new collaborations to obtain these tissues, which we feel would unnecessarily delay publication. We do, however, now include some discussion of the potential caveat of having such a limited samples size for the human comparisons in the revised submission.

4. *The discussion would benefit from more speculation about why the H3 mark goes in opposite directions in resilient mice and susceptible mice between "Day 11" and "Day 40" (data shown in Figs 2/3) post stress and how that may relate to serotonin levels in the DRN.*

Response: We now provide additional text in the Discussion addressing differential regulation of the mark at day 11 vs. day 40 and how alterations in serotonin levels/biosynthesis and/or TGM2 regulation may contribute to the effects observed.

Minor comments:

1. *In line 603, the authors describe 10 days of CSDS as acute and it's a little misleading/confusing.*

Response: We apologize for any confusion with respect to our wording in this instance. We have now clarified this statement in the text.

2. It's not entirely clear why only control group data are shown for FST and EPM (Supp 5), was this a separate cohort of mice that were not defeated?

Response: The data presented in **Fig. S5** are indeed from an independent cohort of non-stressed, virally transduced mice. We have now clarified this information in the text/Figure legend.

3. In line 775, the authors refer to levels of H3 ser being rescued in resilient mice...but is this really a rescue? Is it just that it never went down? I know there are other data suggesting that some CSDS induced changes initially go the same direction in Sus and Res groups then rebound, but this isn't really shown here. Maybe a little more discussion could benefit refinement of this interpretation.

Response: This is an excellent point. We have now adjusted our description of these observations to more accurately reflect the data comparing stress-susceptible vs. stress-resilient mice.

4. It's a bit frustrating that so much data is in the supplements as opposed to main figures, but I appreciate the authors including the thorough characterization of their sequencing data.

Response: We have done our very best to only include the most central pieces of data in the primary Figures, reserving the Supplement for orthogonal experiments and analyses that function to support the primary findings presented within the main Figures. But we appreciate the Referee's recognition of our "thorough characterization of [the] sequencing data."

Reviewer #2:

The manuscript "Histone serotonylation in dorsal raphe nucleus contributes to stress- and antidepressant-mediated gene expression and behavior" uses a combination of RNA sequencing, chromatin immunoprecipitation, mouse behavior, and elegant viral vector tools to demonstrate that stress susceptibility in mice and depression in humans is associated with reduced overall histone serotonylation in the DRN. Surprisingly, they also found that virally decreasing global histone serotonylation in the DRN promotes stress resilience and reverses stress-induced gene expression in mice. Using ChIP sequencing, they demonstrate specific patterns of stress-induced histone serotonylation of genes that are absent in resilient animals and reversed by chronic treatment with the antidepressant fluoxetine. Thus, it seems likely that rather than behavioral effects being driven by global levels of histone serotonylation, they are driven by altered serotonylation levels at histones associated with a specific subset of genes, and that this regulation is prevented by the viral downregulation of histone serotonylation. The authors present a great deal of RNAseq and ChIPseq data that the field can now use to potentially determine what those specific gene targets may be. Thus, this study presents strong evidence for a novel and paradigm-shifting mechanism underlying depressive disorders and the biological functions explaining the effects of current antidepressant treatments.

Overall, this is an extremely well-written paper presenting a great deal of impactful data. The

methods and statistical analyses are appropriate, the figures are clear and present the data in a useful manner, the supplemental data are useful, and the results are interpreted suitably. The introduction and discussion sections set up and interpret the experiments with copious use of the literature in a manner that is clear and readable. I have only a few minor concerns. If these issues are addressed with appropriate text edits, this manuscript will be very suitable for publication and will make an impactful addition to the field that is certain to be cited extensively.

Response: We greatly appreciate the Referee's enthusiasm for our manuscript. We have thoughtfully addressed all of their minor concerns below.

Minor Concerns:

1) The surprising result that stress susceptibility in mice and depression in humans is associated with reduced overall histone serotonylation in the DRN and yet virally decreasing global histone serotonylation in the DRN promotes stress resilience should be addressed more directly in the discussion. Readers unfamiliar with the complexity of gene-behavior interactions may see this as a contradiction, and so the authors should spend some time at the beginning of the discussion section explaining how these results are actually consistent with their interpretations.

Response: This is an excellent point, and we agree that additional discussion regarding this apparent "contradiction" is warranted. We now provide additional discussion of this point at the beginning (third paragraph) of the Discussion section of the manuscript.

2) The data in supplemental figure 5 are quite interesting. They show that reducing histone serotonylation in DRN does not prevent stress-induced anxiety-related behaviors. However, fluoxetine is effective in treating anxiety disorders and anxiety-related symptoms in depression patients. This begs the question of whether histone serotonylation in other brain regions (amygdala?) could contribute to these behaviors, or whether SSRIs work via different mechanisms to affect mood vs anxiety. Experiments to investigate this are beyond the scope of the current study, but some speculation in discussion is warranted.

Response: We fully agree with the points raised above, and while we feel that these studies are indeed beyond the scope of the current manuscript (as pointed out by the Reviewer), we have added additional discussion to the paper (fourth paragraph of the Discussion) in order to highlight the need for future investigations focused on potential differences in patterns of regulation for the mark across brain regions, and their potential roles in regulating other aspects of stress-mediated behavior (e.g., anxiety-related behaviors).

3) The authors should speculate in the discussion on how their novel findings could be leveraged to improve current treatments or drive new treatments for mood disorders. This includes proposing future experiments to uncover the enzymatic mechanisms regulating histone serotonylation in the brain and how these could be pharmacologically manipulated.

Response: Thank you for this suggestion. In accordance with the Reviewer's comment, we have now added further discussion (fourth paragraph of the Discussion) on how these findings could

possibly be leveraged to improve current treatments, or drive new treatments, for mood-related disorders.

Reviewer #3:

Al-Kachak et al. characterized the impact of stress/depression and antidepressant treatment on the status of recently-discovered histone seronylation (H3K4me3Q5ser), associated with transcriptional permissiveness. The authors found that chronic social stress alters H3K4me3Q5ser in the dorsal raphe among both male and female mice, which is rescued by antidepressant treatment. Of particular translational relevance, postmortem human dorsal raphe was also depleted of H3K4me3Q5ser among MDD subjects, but restored with concurrent antidepressant usage. However, in mice, preventing seronylation by expressing a dominant-negative Q5A promotes resilience. This is an interesting and important study, addresses SABV, has taken advantage of cutting-edge techniques and analyses, and is extremely well-written. I have mostly minor comments/questions.

Response: We appreciate this Reviewer's excitement for our manuscript and their feeling that our findings are important for the field. Please find our detailed responses to their critiques below.

Main comments:

1. The original findings shown in Fig2 were that susceptibility/MDD reduced H3K4me3Q5ser, and antidepressants restored levels. However, results presented in Fig3 show the opposite, with susceptibility increasing H3K4me3Q5ser and FLX reducing it. The authors are careful to say "dysregulated/disrupted," and write that this difference is likely due to acute measurements of H3K4me3Q5ser after CSDS originally, vs long-term measurements one month later with H2O/Flx treatment. However, the time scale differences do not explain the directions observed from human brain, where depression and AD treatment are also at longer time scales. The authors speculate in the discussion about intracellular stores and histone turnover rates (again related to time scales), or perhaps brain region specificity (although all conflicting measurements were in DRN), or cause of death. But it seems like some simpler experiments could begin to address this. Does acute SSRI treatment increase Q5ser? Do MAOI's (with distinct monoamine-targeting mechanism) increase or decrease Q5ser with acute vs chronic treatment? The paper deserves to be published even without these experiments or if this is not resolved. It's a great Discussion, truly – the authors dive right in and even use this as a call for further well-controlled animals studies and I almost feel bad harping on this point more here.

Response: We very much thank the Reviewer for their constructive comments in this regard. As discussed above in response to Referees #1 and #2, we now provide additional discussion regarding the differential patterns of H3K4me3Q5ser dysregulation observed between day 11 vs. day 40, including potential mechanisms involved (to be investigated in future efforts) as well as the mark's regulation in the context of chronic FLX treatments. We sincerely hope that such additional discussions will help to clarify our interpretations of the data presented.

Additionally, while we have not yet performed experiments to assess the impact of chronic MAOIs on the mark's regulation in DRN following chronic stress, we did run a pilot experiment (data not shown) demonstrating that acute FLX vs. vehicle (H₂O) treatments do not lead to alterations in H3K4me3Q5ser levels in DRN of stress-susceptible mice. While we have not yet run a full cohort of animals to explore the impact of acute FLX on H3K4me3Q5ser levels in DRN of control vs. stress-susceptible vs. stress-resilient mice to be able to compare to data presented in Fig. 3 (chronic FLX treatments), we posit that since no effects were observed with acute FLX in stress-susceptible mice that such an experiment would result in an unnecessary use of animals to explore a likely negative effect. And given that we do not have data to support a full statistical comparison between acute vs. chronic FLX treatments in all groups of mice, we would respectfully request that such an experiment/data be reserved for follow up studies. However, if the Reviewer and/or Editor felt this data necessary for publication of the current manuscript, then we could present the data we have demonstrating that acute FLX vs. vehicle (H₂O) treatments do not lead to alterations in H3K4me3Q5ser levels in DRN of stress-susceptible mice exclusively in the Supplementary Materials.

Minor comments:

1. *Have other studies found that 5HT itself is reduced among susceptible mice in DRN? Is the (initial) decrease in 5HT binding to H3Q5 a matter of less or depleted 5HT?*

Response: While there are certainly a number of publications that indicate that components of the serotonin system in brain are affected in their expression by social defeat stress (e.g., 5-HT transporter and 5-HT receptors), it remains unclear whether CSDS impacts 5-HT biosynthesis and/or intracellular levels of the monoamine, which may, in turn, contribute to alterations in the mark. And while we have attempted to get at this issue using ELISA-based approaches to monitor total levels of 5-HT in DRN following CSDS, we have not yet observed any differences (data not shown). However, a caveat of such approaches is that the 5-HT being detected is not just the intracellular pool of the monoamine, but rather all 5-HT within the tissues preparations. Ideally, we would like to be able to specifically monitor fluctuations in intracellular 5-HT in response to stress, which is something that we are currently working on using intracellular (cytoplasmic vs. nuclear vs. chromatin) 5-HT sensors. However, these studies are still in their infancy, and we feel that such experiments are beyond the scope of the current manuscript and would be better suited for follow up studies. Having said this, we now provide some extended discussion of possible reasons why the mark may be distinctly regulated during acute vs. protracted periods following chronic stress and provide suggestions for future avenues of inquiry.

2. *Figure 1 A vs D: Why use different behavioral scoring methods for males vs females?*

Response: We appreciate the Reviewer's comment. In brief, sex-specific behavioral effects can often require sex-specific protocols to accurately measure them. Like observations of fear-conditioned female rats (Gruene et al. 2015), socially defeated female mice exhibit active rather than passive avoidance behaviors during social interaction tests. While passive social avoidance is readily measured in socially defeated males using the traditional open field social interaction test, during which time a stimulus animal is held within a small cage, active avoidance in defeated females is most accurately examined using the home cage unrestricted social interaction

test (Newman et al. 2019, 2021, Duque-Wilckens et al. 2017). For this reason, the protocols and behavioral measurements of social deficits following social defeat stress are sex-specific.

3. *There's a lot going on with the colors. I know it's a painful request, but any chance the control males and females could be the same color, and the defeated females could also be blue like susceptible males? That way at least the green/purple scheme used to denote males vs females doesn't overlap with the control/defeat female color scheme (it's confusing to know whether we are looking at sex differences or CvsD females in the venn diagrams in 2, for example, and the Sus circle in H should at least match the Sus male color). Is it also possible to make the Fig 3 sus/res colors match the previous sus/res colors (with the different types of hashing for Post-H2O vs Post-FLX, that part works).*

a. *Are the groups in Fig3D all pre-treatment, based on lack of hashing? Wouldn't they have to be post-treatment? If so, this doesn't appear to match the patterns from 3B-C. I get the use of new colors for 3E-F (and another one in 3H), but... it would be much easier for the reader to connect all of the dots if color was used consistently with the previous parts of the figure. Shade instead of texture could denote treatment.*

b. *Other main and supplemental figures would also benefit from consistency.*

Response: We apologize for any confusion that may have resulted from the color schemes used in our initial submission, and we have done our very best to follow the Reviewer's guidance in adjusting bar plot and Venn diagram colors to be more consistent (and understandable) throughout.

4. *WRT behavior in Fig4C: The lack of a previous statistical difference with H3.3Q5A (particularly given a visual difference between the control and CSDS groups) is not itself significant. Please report the associated p-value and effect size in the legend alongside the other values. Is there a difference between CSDS WT and CSDS Q5A? Even if these numbers are less convincing, the RRHO's are still very convincing.*

Response: Data for this Figure (Now Fig. 4D) were analyzed using a two-way ANOVA, with significant main effects of stress observed ($p=0.0001$, $F_{1,57} = 17.29$). Bonferroni's multiple comparisons tests revealed significant differences between control vs. CSDS groups in GFP ($p=0.0310$) and H3.3 WT mice ($p=0.0474$), with no differences observed between control vs. CSDS H3.3Q5A mice. These statistical results are now reported, as requested, in the legend. Using the same two-way ANOVA but with a Tukey's multiple comparison test to determine cross virus effects did not yield significant differences (e.g., CSDS WT vs. CSDS H3.3Q5A, $p=0.1190$), and as such, they are not reported in the manuscript. However, we have adjusted our descriptions of these data in the text to make it clear how the comparisons presented were made.

5. *Supp Fig2: Why are only the control genomic compartment %s shown? Did stress/susceptibility alter these distributions at all?*

Response: In the manuscript, we only show the control genomic compartment %s (based upon peak calling analyses) to provide the reader with an overall view of where the H3K4me3Q5ser mark is normally enriched. And since stress does not appreciably affect global patterns of compartmental localization for the mark or the total number of peaks called, we chose to not

include what would essentially be a repeat of the control data in the Supplementary Figure. However, while we did not observe significant changes in the total number, or location, of peaks called, we did identify robust alterations in the mark's enrichment at many genomic loci as a consequence of stress, data which are provided throughout the manuscript. And since in our analysis pipeline, we required that peaks be called for a given protein-coding gene in at least one of the sample groups being assessed per experiment for it to be considered differentially enriched by stress, our analyses are focused on loci that normally display enrichment for the mark and are subsequently altered as a consequence of the stress exposures.

6. Supp3: Recommend increasing the RRHO scale.

Response: This has been corrected as recommended by the Reviewer.

Reviewers' Comments:

Reviewer #1:

Remarks to the Author:

The authors have been very responsive to my critique as well as the other reviewers. In particular, my biggest concern has been addressed with the new immuno data in Fig. 4 demonstrating a significant knockdown of H3K4me3Q5ser in HA+ cells. I find this work acceptable and look forward to its publication.

Reviewer #2:

Remarks to the Author:

The authors did an outstanding job of addressing all concerns. Looking forward to seeing this exciting manuscript in print!

Reviewer #3:

Remarks to the Author:

The authors have thoroughly addressed my concerns.

REVIEWERS' COMMENTS

Reviewer #1 (Remarks to the Author):

The authors have been very responsive to my critique as well as the other reviewers. In particular, my biggest concern has been addressed with the new immuno data in Fig. 4 demonstrating a significant knockdown of H3K4me3Q5ser in HA+ cells. I find this work acceptable and look forward to its publication.

Reviewer #2 (Remarks to the Author):

The authors did an outstanding job of addressing all concerns. Looking forward to seeing this exciting manuscript in print!

Reviewer #3 (Remarks to the Author):

The authors have thoroughly addressed my concerns.

*We very much thank all three Reviewer's for their continued enthusiasm for our manuscript, and their feeling that this work is now suitable for publication at *Nature Communications**